# Advantage Collapse in Group Relative Policy Optimization: Diagnosis and Mitigation

**Xixiang He**[1]  **Qiyao Sun**[1]  **Ao Cheng**[1]  **Xingming Li**[1]  **Xuanyu Ji**[1]  **Hailun Lu**[2]  **Runke Huang**[3]  **Qingyong Hu**[2]

## Abstract

Group Relative Policy Optimization (GRPO), a prominent algorithm within the Reinforcement Learning from Verifiable Rewards (RLVR) framework, has achieved strong results in improving the reasoning capabilities of large language models (LLMs). However, GRPO is prone to *advantage collapse*, a failure mode where homogeneous rewards within a group (*e.g.*, all correct or all incorrect answers) yield near-zero advantages and vanishing gradients. To address this, we introduce the Advantage Collapse Rate (ACR), the first diagnostic metric quantifying the proportion of training batches with ineffective gradients. Across models from 0.5B to 14B parameters on mathematical reasoning benchmarks, we show that ACR strongly predicts training stagnation and final performance. We then propose Adaptive Virtual Sample Policy Optimization (AVSPO), a lightweight extension of GRPO that injects virtual reward samples, guided by real-time ACR monitoring, to enable learning from homogeneous groups without additional model rollouts. AVSPO reduces advantage collapse by 58–63% relative to GRPO and yields consistent accuracy gains of 4–6 percentage points across all model scales, while maintaining generalization on the evaluated out-of-domain task. Code and datasets are available at https://github.com/hexixiang/Advantage-Collapse-Rate.

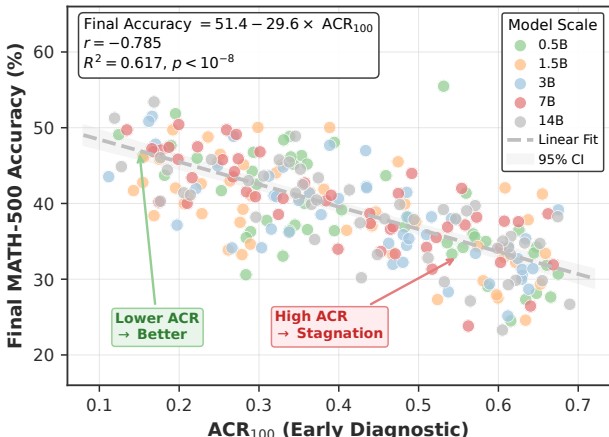

*Figure 1.* **Early ACR predicts final performance.** Each point represents one training run across 245 configurations (5 model scales, 7 difficulty levels, 7 group sizes). Strong negative correlation ($R^2 = 0.617$) between early-stage ACR and final accuracy. Points colored by model scale.

## 1. Introduction

Reinforcement learning from verifiable rewards (RLVR) has emerged as a promising paradigm for improving mathematical reasoning in large language models (Shao et al., 2024; Guo et al., 2025). Unlike Reinforcement Learning from Human Feedback (RLHF) (Ouyang et al., 2022), which relies on human preference, RLVR leverages automated verifiers to provide binary feedback on solution correctness. Within this framework, Group Relative Policy Optimization (GRPO) has emerged as a leading approach. GRPO computes advantages through intra-group reward comparisons rather than learned value baselines. This design eliminates the critic network required by actor-critic methods such as PPO (Schulman et al., 2017), significantly reducing memory consumption.

However, GRPO suffers from a known failure mode termed *advantage collapse* (Zhang et al., 2025a; Zhang & Zuo, 2025; Yu et al., 2026; Le et al., 2025). When all responses in a group receive identical rewards, the within-group vari-

---

[1]National University of Defense Technology, Changsha, Hunan, China [2]Intelligent Game and Decision Lab, Beijing, China [3]The Chinese University of Hong Kong, Shenzhen, Guangdong, China. Correspondence to: Xixiang He <hexixiang@nudt.edu.cn>, Qingyong Hu <huqingyong15@outlook.com>.

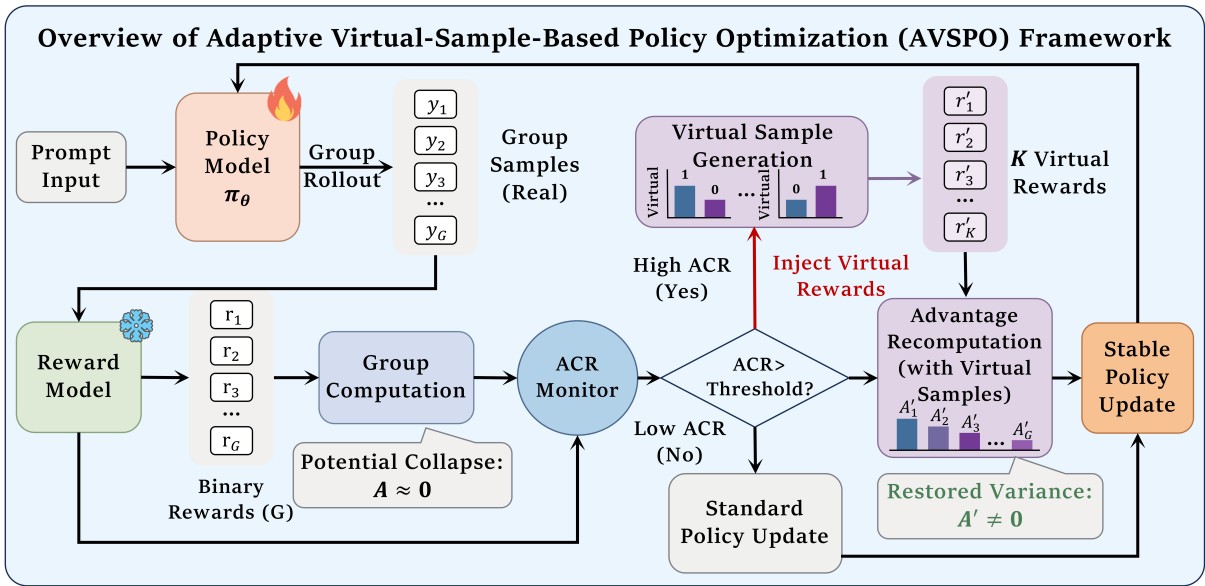

*Figure 2.* **Overview of AVSPO.** The policy generates $G$ responses per prompt with binary rewards. When all rewards are identical, GRPO's advantages collapse to zero ($A = 0$). AVSPO monitors ACR in real-time; upon detecting collapse, it injects virtual samples to restore reward variance and recompute non-zero advantages ($A' \neq 0$) for stable policy updates.

ance vanishes, causing all advantages to collapse to zero. The policy gradient then becomes ineffective regardless of individual response quality. This phenomenon is prevalent in mathematical reasoning tasks where binary verification often yields homogeneous outcomes. In our experiments across five model scales on mathematical reasoning benchmarks, we observe that 28–45% of training batches exhibit complete advantage collapse. This failure remains invisible to conventional metrics such as loss curves and accuracy, leading to wasted computation.

Several strategies have been proposed to address gradient ineffectiveness in policy optimization. Generalized Advantage Estimation (GAE) (Schulman et al., 2015) provides variance reduction but requires a critic network incompatible with GRPO's architecture. Process Reward Models (PRMs) (Lightman et al., 2024) offer dense supervision but demand expensive step-level annotations. Entropy regularization (Shen, 2025) encourages exploration but applies uniform pressure regardless of reward structure. Recent GRPO variants such as DAPO (Yu et al., 2026) introduce system-level improvements for long-CoT RL, including clipping modifications and dynamic sampling, yet none provides a diagnostic tool to quantify collapse severity before it impacts training.

This gap motivates a different perspective: rather than modifying the optimization algorithm, we focus on *diagnosing* advantage collapse in real time. Our key insight is that collapse is directly measurable through reward variance statistics already computed during GRPO training. As shown

in Figure 1, early-stage collapse rate strongly predicts final performance ($R^2 = 0.617$), enabling early detection of problematic configurations before substantial compute is wasted. This predictive power motivates our approach: real-time monitoring combined with adaptive intervention when collapse is detected. To summarize, our contributions are threefold:

- **Diagnostic metric.** We introduce the *Advantage Collapse Rate* (ACR), the first metric specifically designed to quantify gradient ineffectiveness in group-based policy optimization. ACR leverages statistics already computed in GRPO. Early-stage ACR explains 62% of variance in final performance, enabling practitioners to identify problematic configurations before accuracy degradation becomes visible.

- **Algorithmic intervention.** We propose *Adaptive Virtual Sample Policy Optimization* (AVSPO), a plug-and-play extension of GRPO that injects virtual reward samples when collapse is detected. AVSPO creates non-zero advantages to enable learning from otherwise wasted batches, requiring no additional LLM forward passes.

- **Empirical validation.** Across models from 0.5B to 14B parameters on six mathematical reasoning benchmarks, AVSPO reduces advantage collapse from 28–45% to 11–18% (58–63% relative reduction), yielding accuracy gains of 4–6 percentage points over GRPO

while maintaining generalization on the out-of-domain benchmark (MMLU-Pro).

Our diagnostic approach complements algorithmic improvements. ACR can monitor any group-based policy optimization method, beyond GRPO. We hope this work motivates the community to develop real-time diagnostics for training efficiency, enabling practitioners to detect and address gradient ineffectiveness before wasting computation.

## 2. Related Work

**GRPO and Advantage Collapse.** GRPO (Shao et al., 2024) eliminates the critic network required by PPO (Schulman et al., 2017), significantly reducing memory consumption (Guo et al., 2025; Lambert et al., 2024). However, identical within-group rewards cause advantage collapse (Zhang & Zuo, 2025; Yu et al., 2026; Le et al., 2025). Existing solutions fall short: GAE (Schulman et al., 2015) requires a critic; PRMs (Lightman et al., 2024) need expensive annotations; entropy regularization (Shen, 2025; Deng et al., 2025) ignores reward structure.

**GRPO Variants.** Prior works address GRPO limitations at the *optimization level*: gradient estimation (Liu et al., 2025; He et al., 2025; Hu et al., 2025), clipping mechanisms (Yang et al., 2025a), system-level recipes with dynamic sampling (Yu et al., 2026), value estimation (Sane, 2025), and token-level weighting for sharpness control (Le et al., 2026). Others operate at the *policy level*: reward shaping (Gupta et al., 2025), exploration (Nan et al., 2025; Hou et al., 2025), and entropy control (Wang et al., 2025). Concurrent works target related failure modes (Li et al., 2025; Bai et al., 2025; Yang et al., 2026); Scaf-GRPO (Zhang et al., 2025b) addresses collapse via progressive in-prompt scaffolding. AVSPO instead intervenes at the *reward statistics level*, making it complementary.

**Diagnostic Metrics.** Standard RL monitoring uses policy entropy, KL divergence, and gradient norms (Castanyer et al., 2025; Henderson et al., 2018; Zheng et al., 2025). Parallel efforts target reward hacking detection (Shihab et al., 2025) and entropy-driven instability in off-policy settings (Xi et al., 2025). Recent work (Yang et al., 2025b) identifies token-level imbalances but not batch-level collapse. Existing diagnostics assume critic-based methods (Mroueh, 2025). ACR fills this gap by quantifying gradient ineffectiveness for GRPO.

## 3. Preliminaries

We frame mathematical reasoning as a contextual bandit problem (Langford & Zhang, 2007) under the RLVR paradigm. Given a question $q \sim \mathcal{Q}$, a language model $\pi_\theta$ autoregressively generates a solution trajectory $y =$ $(y_1, \ldots, y_T)$. Upon completion, an automated verifier $V$ (*e.g.*, via code execution or symbolic matching) assigns a binary reward $r(q, y) \in \{0, 1\}$. The objective is to maximize expected reward:

$$J(\theta) = \mathbb{E}_{q \sim \mathcal{Q},\, y \sim \pi_\theta(\cdot|q)}[r(q, y)]. \tag{1}$$

This sparse, outcome-only supervision poses a fundamental challenge: how to extract effective learning signals from binary feedback (Riedmiller et al., 2018).

### 3.1. Group Relative Policy Optimization

To address this while maintaining scalability, GRPO eliminates the critic network and instead computes advantages via *intra-group comparisons*. For each question $q$, it samples $G$ independent responses $\mathcal{O} = \{y^{(1)}, \ldots, y^{(G)}\}$ from the old policy $\pi_{\theta_{\text{old}}}$ and obtains their rewards $\mathcal{R} = \{r_1, \ldots, r_G\}$. The advantage for the $i$-th sample is:

$$\hat{A}_i = \frac{r_i - \mu_\mathcal{R}}{\sigma_\mathcal{R} + \epsilon}, \tag{2}$$

where $\mu_\mathcal{R}$ and $\sigma_\mathcal{R}$ are the group mean and standard deviation, and $\epsilon > 0$ ensures numerical stability. The policy is updated via a clipped surrogate objective:

$$\mathcal{J}^{\text{GRPO}}(\theta) = \mathbb{E}_{q \sim \mathcal{Q},\, \mathcal{O} \sim \pi_{\theta_{\text{old}}}(\cdot|q)} \left[ \frac{1}{G} \sum_{i=1}^{G} \frac{1}{|y^{(i)}|} \sum_{t=1}^{|y^{(i)}|} \right.$$
$$\left. \min\left( \rho_t^{(i)} \hat{A}_i,\ \text{clip}(\rho_t^{(i)}, 1-\varepsilon, 1+\varepsilon) \hat{A}_i \right) \right], \tag{3}$$

where $\rho_t^{(i)} = \pi_\theta(y_t^{(i)}|q, y_{<t}^{(i)}) / \pi_{\theta_{\text{old}}}(y_t^{(i)}|q, y_{<t}^{(i)})$ is the policy ratio, and $\varepsilon$ is the clipping range (distinct from $\epsilon$ for numerical stability). We set the KL penalty $\beta = 0$ following standard practice (Shao et al., 2024).

### 3.2. The Advantage Collapse Phenomenon

Despite its efficiency, GRPO exhibits a fundamental failure mode under reward homogeneity. Consider a group where all samples receive identical rewards $r_i = c$ for some constant $c \in \{0, 1\}$. Then $\mu_\mathcal{R} = c$ and $\sigma_\mathcal{R} = 0$, yielding:

$$\hat{A}_i = \frac{c - c}{0 + \epsilon} = 0, \quad \forall i \in [G]. \tag{4}$$

We term this **advantage collapse**: the complete loss of gradient signal when within-group reward variance vanishes. The policy gradient contribution from such a group becomes:

$$\sum_{i=1}^{G} \hat{A}_i \nabla_\theta \log \pi_\theta(y^{(i)}|q) = \mathbf{0}. \tag{5}$$

This pathology manifests in two symmetric scenarios under binary rewards:

- **All-incorrect** ($r_i = 0 \; \forall i$): The model fails uniformly on hard problems.

- **All-correct** ($r_i = 1 \; \forall i$): The model succeeds uniformly on easy problems.

In both cases, the verifier provides valid feedback, yet GRPO extracts no learning signal. This failure is *invisible* to standard diagnostics: loss curves and accuracy metrics appear stable while gradient updates become ineffective. Addressing this requires a diagnostic to detect collapse and an intervention to recover collapsed groups, motivating our ACR metric and AVSPO algorithm (Section 4).

# 4. Methodology

We present a unified approach to diagnosing and mitigating advantage collapse in GRPO. Our method comprises two tightly integrated components: (1) the **Advantage Collapse Rate (ACR)**, a lightweight diagnostic metric that quantifies gradient ineffectiveness in real-time, and (2) **Adaptive Virtual Sample Policy Optimization (AVSPO)**, an ACR-guided intervention mechanism that enables learning from collapsed groups. Together, these components form a principled framework for stabilizing GRPO training without compromising its computational efficiency.

## 4.1. Advantage Collapse Rate (ACR)

ACR operationalizes the collapse condition inherent in GRPO's advantage formula (Eq. 2). Recall that $\hat{A}_i \propto (r_i - \mu_{\mathcal{R}})/\sigma_{\mathcal{R}}$. When $\sigma_{\mathcal{R}} \to 0$, advantages vanish regardless of individual reward magnitudes, causing gradients to collapse. ACR directly monitors this condition.

**Definition 4.1** (Advantage Collapse Rate). Given a training batch consisting of $N$ question-group pairs $\{(q_1, \mathcal{O}_1), \ldots, (q_N, \mathcal{O}_N)\}$, where each group $\mathcal{O}_j = \{y_j^{(1)}, \ldots, y_j^{(G)}\}$ contains $G$ sampled solutions with corresponding rewards $\mathcal{R}_j = \{r_j^{(1)}, \ldots, r_j^{(G)}\}$, ACR is defined as:

$$\text{ACR} = \frac{1}{N} \sum_{j=1}^{N} \mathbb{I} \left( \sigma_{\mathcal{R}_j} < \tau \right) \quad (6)$$

where $\sigma_{\mathcal{R}_j} = \sqrt{\frac{1}{G} \sum_{i=1}^{G} (r_j^{(i)} - \mu_{\mathcal{R}_j})^2}$ is the standard deviation of rewards in group $j$, $\mathbb{I}(\cdot)$ is the indicator function, and $\tau$ is a small threshold (typically $\tau = 10^{-6}$) accounting for numerical precision.

**Interpretation.** ACR quantifies the proportion of groups within a batch that have negligible gradient signals:

- **ACR $\approx$ 0**: Optimal learning conditions. All groups exhibit reward diversity, ensuring non-zero gradients throughout the batch.

- **ACR $\approx$ 1**: Complete gradient stagnation. Every group suffers from reward homogeneity, causing vanishing learning signals.

- $0 < $ **ACR** $< 1$: Partial effectiveness. The magnitude indicates the fraction of computational resources spent on ineffective gradient updates.

ACR incurs zero computational overhead since it monitors reward statistics already computed during standard GRPO training.

## 4.2. Adaptive Virtual Sample Policy Optimization

When ACR indicates advantage collapse ($\text{ACR}^{(n)} > \tau_{\text{adapt}}^{(n)}$), AVSPO generates virtual samples to restore reward diversity. Virtual samples are *synthetic reward values*, not actual model outputs, that participate only in the normalization statistics ($\mu_{\mathcal{R}'}$ and $\sigma_{\mathcal{R}'}$) for advantage computation. They do not contribute to the policy gradient, as no corresponding $\nabla_\theta \log \pi_\theta$ term exists for virtual samples.

**Construction Mechanism.** For a collapsed group $\mathcal{O}_j$ with homogeneous rewards $\mathcal{R}_j$, we construct a set of virtual samples:

$$\mathcal{V}_j = \{v_1, v_2, \ldots, v_K\} \quad (7)$$

where the number of virtual samples $K$ is determined adaptively:

$$K = \max \left( 1, \min \left( G, \left\lceil G \cdot (\text{ACR}^{(n)})^\alpha \right\rceil \right) \right) \quad (8)$$

with sensitivity parameter $\alpha \in (0, 1]$ controlling augmentation strength. Setting $\alpha = 0.5$ provides effective scaling: when ACR is low ($\approx 0$), minimal intervention ($K \approx 1$); when ACR is high ($\approx 1$), stronger support ($K \approx G$).

**Stratified Reward Assignment.** Each virtual sample $v_k$ is assigned a reward value that ensures non-zero variance regardless of whether the collapsed group consists of all-correct or all-incorrect responses. Let $r_{\text{obs}} = \max(\mathcal{R}_j)$ denote the observed maximum reward in the original group. The virtual rewards are assigned as:

$$r_{v_k} = \begin{cases} r_{\text{obs}} \cdot \left( 1 - \frac{k}{K+1} \right) & \text{if } r_{\text{obs}} > 0 \\ r_{\text{anchor}} \cdot \frac{K-k+1}{K} & \text{if } r_{\text{obs}} = 0 \end{cases} \quad (9)$$

where $r_{\text{anchor}} > 0$ is a small positive constant (we use $r_{\text{anchor}} = 0.1$ in all experiments). Note that while actual rewards are binary ($r \in \{0, 1\}$), virtual rewards take continuous values in $(0, 1]$ to create reward diversity for normalization. The denominator ($K + 1$) in the first case ensures that even with minimal intervention ($K = 1$), the virtual reward creates sufficient contrast with the original homogeneous rewards, guaranteeing $\sigma_{\mathcal{R}'_j} > 0$.

The critical distinction is the handling of **all-incorrect groups** ($r_{\text{obs}} = 0$), which dominate early training in sparse-reward tasks. Our formulation introduces positive anchor rewards that create contrast against the zero rewards of actual samples, ensuring $\sigma_{\mathcal{R}'_j} > 0$. The constraint $K \leq G$ ensures virtual samples never outnumber real ones, limiting the bias introduced into advantage estimates (analyzed in Section 4.3).

**Adaptive Triggering Mechanism.** AVSPO employs dynamic thresholding that adapts the intervention trigger based on training progress, avoiding both under-intervention (missing collapse) and over-intervention (unnecessary augmentation).

**Real-time ACR Monitoring.** At each training iteration $n$, ACR is computed over the current batch using Equation 6, denoted as $\text{ACR}^{(n)}$.

**Dynamic Threshold Adaptation.** The augmentation threshold $\tau_{\text{adapt}}$ is initialized conservatively ($\tau_{\text{adapt}}^{(0)} = 0.5$) and adjusted based on training stability:

$$\tau_{\text{adapt}}^{(n+1)} = \tau_{\text{adapt}}^{(n)} + \eta \cdot \text{sign}\left(\Delta J^{(n)}\right) \cdot \left(\text{ACR}^{(n)} - \tau_{\text{adapt}}^{(n)}\right) \tag{10}$$

where $\Delta J^{(n)} = \hat{J}(\theta^{(n)}) - \hat{J}(\theta^{(n-1)})$ measures policy improvement estimated via average batch reward $\hat{J}^{(n)} = \frac{1}{NG} \sum_{j,i} r_j^{(i)}$, and $\eta = 0.01$ is a small learning rate. **Intuition:** When training improves ($\Delta J > 0$), $\tau_{\text{adapt}}$ tracks the observed ACR, reducing unnecessary intervention; when training stagnates ($\Delta J < 0$) under high ACR, $\tau_{\text{adapt}}$ decreases to trigger stronger correction.

**Conditional Sample Integration.** Virtual samples are incorporated only when collapse is detected:

$$\mathcal{R}'_j = \begin{cases} \mathcal{R}_j \cup \mathcal{V}_j & \text{if } \text{ACR}^{(n)} > \tau_{\text{adapt}}^{(n)} \text{ and } \sigma_{\mathcal{R}_j} < \tau \\ \mathcal{R}_j & \text{otherwise} \end{cases} \tag{11}$$

**Advantage Recomputation.** With the augmented reward set $\mathcal{R}'_j$, advantages are recomputed to incorporate increased reward diversity:

$$\hat{A}_j^{(i)\prime} = \frac{r_j^{(i)} - \mu_{\mathcal{R}'_j}}{\sigma_{\mathcal{R}'_j} + \epsilon} \tag{12}$$

where $\mu_{\mathcal{R}'_j} = \frac{1}{G+K}\left(\sum_{i=1}^{G} r_j^{(i)} + \sum_{k=1}^{K} r_{v_k}\right)$ and $\sigma_{\mathcal{R}'_j}$ is the standard deviation of $\mathcal{R}'_j$. Note that advantages are computed only for the $G$ *real* samples; virtual rewards serve solely to shift the normalization baseline, ensuring $\sigma_{\mathcal{R}'_j} > 0$ even when $\sigma_{\mathcal{R}_j} = 0$. The final AVSPO objective function becomes:

$$\mathcal{J}^{\text{AVSPO}}(\theta) = \mathbb{E}\left[\frac{1}{G}\sum_{i=1}^{G}\frac{1}{|y^{(i)}|}\sum_{t=1}^{|y^{(i)}|}\right.$$
$$\left.\min\left(\rho_t^{(i)}\hat{A}'_i, \text{clip}\left(\rho_t^{(i)}, 1-\varepsilon, 1+\varepsilon\right)\hat{A}'_i\right)\right], \tag{13}$$

where the expectation is over $q \sim \mathcal{Q}$ and $\mathcal{O} \sim \pi_{\theta_{\text{old}}}(\cdot|q)$, and $\hat{A}'_i \equiv \hat{A}_j^{(i)\prime}$ is computed using Equation 12 when augmentation is triggered (with the group index $j$ suppressed for clarity), reducing to the standard GRPO advantage otherwise.

### 4.3. Theoretical Analysis

**Proposition 4.2** (Reward Variance and Gradient Scaling). *For a group $\mathcal{O}$ with rewards $\mathcal{R} = \{r_1, \ldots, r_G\}$ having mean $\mu_{\mathcal{R}}$ and standard deviation $\sigma_{\mathcal{R}}$, the sum of squared advantages satisfies:*

$$\sum_{i=1}^{G} \hat{A}_i^2 = \frac{G\sigma_{\mathcal{R}}^2}{(\sigma_{\mathcal{R}} + \epsilon)^2} \tag{14}$$

*where $\epsilon > 0$ is the numerical stability constant. Consequently, as $\sigma_{\mathcal{R}} \to 0$, we have $\sum_i \hat{A}_i^2 \to 0$, and the contribution of this group to the policy gradient vanishes. (Proof in Appendix B.)*

The quadratic scaling justifies ACR as a diagnostic for gradient ineffectiveness.

**Proposition 4.3** (Bias from Virtual Augmentation). *Let $\nabla_\theta \mathcal{J}^{GRPO}$ denote the standard GRPO gradient estimator and $\nabla_\theta \mathcal{J}^{AVSPO}$ denote the AVSPO gradient. For a collapsed group with $K$ virtual samples and a given realization of sampled responses, the gradient difference is:*

$$\nabla_\theta \mathcal{J}^{AVSPO} - \nabla_\theta \mathcal{J}^{GRPO} = \frac{1}{G}\sum_{i=1}^{G}(\hat{A}'_i - \hat{A}_i)\nabla_\theta \log \pi_\theta(y^{(i)}|q) \tag{15}$$

*For collapsed groups where $\sigma_{\mathcal{R}} = 0$, we have $\hat{A}_i = 0$, thus $|\hat{A}'_i - \hat{A}_i| = |\hat{A}'_i|$. The augmented advantages $|\hat{A}'_i|$ are bounded since $|\hat{A}'_i| \leq |r_i - \mu_{\mathcal{R}'}|/\sigma_{\mathcal{R}'} \leq \sqrt{G+K}$ by properties of standardized variables.*

For collapsed groups, AVSPO enables policy updates at the cost of introducing bias relative to the original gradients. This bias-variance tradeoff (Greensmith et al., 2004) is favorable when the alternative is gradient stagnation.

The complete AVSPO training procedure is provided in Algorithm 1 (Appendix C). The algorithm integrates ACR monitoring with conditional virtual sample injection, requiring only minimal additions to standard GRPO training

loops. AVSPO incurs no additional LLM forward passes and maintains $O(NG)$ per-iteration complexity, identical to vanilla GRPO.

# 5. Experiments

We conduct comprehensive experiments to address three questions: (1) Does AVSPO effectively mitigate advantage collapse and improve reasoning performance? (2) Is ACR a reliable diagnostic for training efficiency? (3) How do individual design choices contribute to performance?

## 5.1. Experimental Setup

**Training Data.** We construct Level3-500 by sampling 500 intermediate-difficulty problems from the MATH training split (Hendrycks et al., 2021), targeting the optimal learning zone for gradient signal diversity (Bengio et al., 2009; Gao et al., 2025). Details on difficulty-based selection are provided in Appendix A.1.

**Evaluation Benchmarks.** We evaluate on six mathematical reasoning benchmarks spanning diverse difficulty levels: MATH-500 (Hendrycks et al., 2021), GSM8K (Cobbe et al., 2021), Minerva (Lewkowycz et al., 2022), Olympiad-Bench (He et al., 2024), AMC, and AIME24, covering elementary arithmetic to Olympiad-level competitions. To assess out-of-domain generalization, we additionally evaluate on MMLU-Pro (Wang et al., 2024), a challenging multi-task language understanding benchmark that tests broader reasoning capabilities beyond mathematics.

**Models.** We experiment with six Qwen2.5 models (Yang et al., 2024) spanning 0.5B to 14B parameters: Qwen2.5-0.5B, Qwen2.5-3B, Qwen2.5-3B-Instruct, Qwen2.5-Math-1.5B, Qwen2.5-Math-7B, and Qwen2.5-14B. All models are trained with group size $G = 8$ and sampling temperature $T = 1.0$. Training is conducted on $8 \times$NVIDIA A800-80GB GPUs using the TRL framework (von Werra et al., 2020), with one problem sampled per GPU per step. Complete hyperparameter settings are provided in Appendix A.

**Baselines.** We compare against: (1) **Base Model** (pretrained without RL); (2) **Vanilla GRPO** (Shao et al., 2024) with standard intra-group normalization; (3) **DCPO** (Yang et al., 2025a), using dynamic clipping and smooth advantage standardization; (4) **INTUITOR** (Zhao et al., 2025), an RLIF approach using self-certainty as reward; and (5) **RENT** (Prabhudesai et al., 2025), minimizing output entropy.

**Evaluation Protocol.** All benchmarks are evaluated using the lighteval library (Habib et al., 2023). We report pass@1 accuracy using greedy decoding ($T = 0$). For competition benchmarks with limited problems (AMC: 25, AIME24: 30), we use avg@32 following Shao et al. (2024) to reduce variance from small sample sizes. Final answers are extracted via rule-based parsing and verified symbolically.

**Training and Reproducibility.** All models are trained for 500 steps. Multi-seed experiments ($n = 5$) on representative configurations confirm statistical significance ($p < 0.05$) for all AVSPO improvements (Appendix I).

## 5.2. Main Results

Table 1 presents results across all configurations.

**Main Findings.** AVSPO reduces ACR from 0.28–0.45 (GRPO) to 0.11–0.18, a 58–63% relative reduction. This translates to +4 to +6 accuracy points over GRPO. Models with higher baseline ACR benefit more: Qwen2.5-Math-1.5B (ACR=0.40) shows the largest gain (+6.1%), consistent with Proposition 4.2.

**Comparison with Baselines.** AVSPO outperforms DCPO (+2.9%), INTUITOR (+6.8%), and RENT (+8.9%). INTUITOR uses internal confidence as the reward signal; RENT minimizes output entropy to encourage decisive reasoning. Both modify the reward signal but do not address advantage diversity. DCPO targets policy divergence through dynamic clipping rather than reward diversity. These results suggest that restoring batch-level gradient signal is more effective than reward engineering given verification.

**Analysis Across Scales and Difficulty.** The improvement pattern varies systematically. On moderate-difficulty benchmarks (GSM8K, MATH-500), AVSPO achieves the largest gains. On competition-level problems (AMC, AIME), gains are more modest, suggesting model capacity becomes the limiting factor. Across model scales, larger models have lower baseline ACR (0.21 for 14B vs. 0.45 for 0.5B) and show smaller but consistent improvements. The correlation between ACR reduction and accuracy gains (Figure 1) indicates that advantage collapse is a primary bottleneck in GRPO training.

## 5.3. Validating ACR as a Diagnostic Metric

To establish ACR as a reliable diagnostic beyond its role in AVSPO, we conduct systematic validation across factors that influence reward diversity: problem difficulty, model capacity, sampling temperature, and group size.

### 5.3.1. ACR PREDICTS FINAL PERFORMANCE

We examine whether early-stage ACR measurements can forecast training outcomes. Across 245 setups (5 model scales $\times$ 7 difficulty levels $\times$ 7 group sizes), we compute the mean ACR over the first 100 training steps ($\mathrm{ACR}_{100}$) and correlate it with final accuracy on MATH-500. As shown in Figure 1, the analysis reveals a strong negative correlation ($r = -0.785$, $p < 10^{-8}$). An ordinary least squares fit

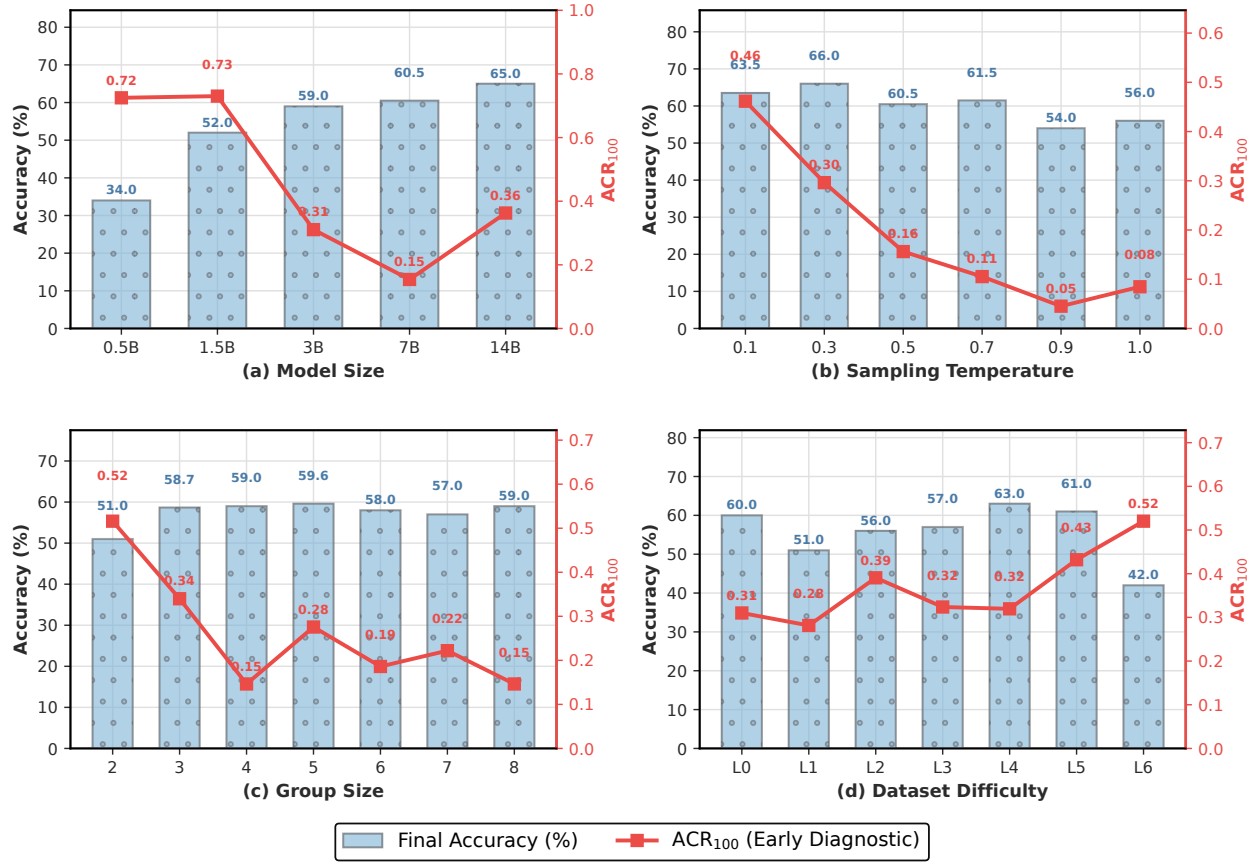

*Figure 3.* **Sensitivity analysis of ACR across training conditions.** Blue bars show final accuracy (left axis); red line shows $ACR_{100}$ (right axis). (a) Larger models generally achieve lower ACR with higher accuracy, though not strictly monotonic. (b) Moderate temperatures ($T = 0.3–0.5$) balance exploration and precision. (c) Increasing group size $G$ reduces ACR with diminishing returns. (d) Intermediate difficulty (L3–L4) yields optimal learning conditions. Note: ACR values here differ from Table 1 as each subplot varies one hyperparameter at non-standard values for sensitivity analysis.

yields:

$$\text{Final Accuracy} = 51.4 - 29.6 \times ACR_{100} \qquad (16)$$

with $R^2 = 0.617$. Thus, $ACR_{100}$ explains 62% of final performance variance, enabling early detection of poor configurations.

### 5.3.2. SENSITIVITY TO TRAINING CONDITIONS

We investigate how ACR responds to four key factors: model capacity, sampling temperature, group size, and problem difficulty (Figure 3). Key findings: (1) larger models achieve lower ACR, though the 7B model outperforms 14B due to task-model matching effects; (2) temperature exhibits an optimal range at $T = 0.3–0.5$, where *low ACR is necessary but not sufficient*—balancing exploration and exploitation; (3) group size $G \in [6, 8]$ provides the best trade-off between gradient effectiveness and computational cost; (4) problem difficulty shows a U-shaped relationship with ACR,

as both easy and hard problems cause uniform rewards, validating our choice of intermediate-difficulty training data. These results establish ACR as a reliable real-time diagnostic with zero computational overhead. Detailed per-factor analysis is provided in Appendix F.

### 5.4. Ablation Studies

We ablate key design choices on Qwen2.5-Math-1.5B to validate AVSPO's components. Table 2 summarizes the virtual sample construction strategies.

**Virtual Sample Construction.** We compare five strategies: (1) no augmentation (GRPO baseline); (2) random uniform sampling $r_v \sim U[0, r_{\max}]$; (3) fixed partial credit ($r_v = 0.5 \cdot r_{\max}$); (4) exponential decay ($r_v^k = r_{\max} e^{-0.5k}$); and (5) stratified assignment (ours). As shown in Table 2, random sampling reduces ACR but introduces variance, yielding only +3.5% improvement. Fixed partial credit achieves +4.9% but fails to span the reward spectrum.

*Table 1.* Performance comparison across model scales and benchmarks. All methods trained for one epoch (500 steps) on Level3-500 with standard configuration ($G$=8, $T$=1.0) and evaluated using greedy decoding. We report pass@1 accuracy (%) and average ACR (lower is better). Note: ACR values in Figure 3 differ as they are computed under varied hyperparameter settings for sensitivity analysis. MMLU-Pro serves as out-of-domain evaluation to assess generalization beyond mathematical reasoning. Avg. is computed over all seven benchmarks. Best results per model in **bold**. Multi-seed validation ($n$=5) on representative configurations (4 models $\times$ 2 benchmarks) confirms statistical significance (Appendix I).

| Model / Method | ACR↓ | MATH | GSM8K | Minerva | Olympiad | AMC[‡] | AIME[‡] | MMLU-Pro | Avg. |
|---|---|---|---|---|---|---|---|---|---|
| *General Models* | | | | | | | | | |
| Qwen2.5-0.5B[Base] | – | 6.8 | 41.1 | 4.1 | 4.8 | 5.2 | 0.0 | 15.5 | 11.1 |
| + INTUITOR (Zhao et al., 2025) | – | 21.5 | 28.4 | 7.3 | 10.2 | 12.1 | 1.2 | 14.9 | 13.7 |
| + RENT (Prabhudesai et al., 2025) | – | 19.8 | 24.1 | 6.2 | 8.9 | 10.5 | 0.8 | 14.6 | 12.1 |
| + DCPO (Yang et al., 2025a) | 0.38 | 26.3 | 37.8 | 10.5 | 14.2 | 16.8 | 2.4 | 15.6 | 17.7 |
| + GRPO (Shao et al., 2024) | 0.45 | 24.6 | 35.2 | 9.8 | 13.5 | 15.3 | 2.1 | 15.1 | 16.5 |
| + AVSPO (Ours) | **0.18** | **31.4** | **44.8** | **13.2** | **17.9** | **20.1** | **3.5** | **16.0** | **21.0** |
| Qwen2.5-3B[Base] | – | 28.0 | 66.3 | 7.4 | 7.5 | 9.0 | 0.3 | 37.9 | 22.3 |
| + INTUITOR (Zhao et al., 2025) | – | 32.4 | 45.8 | 12.6 | 18.7 | 21.3 | 4.1 | 37.1 | 24.6 |
| + RENT (Prabhudesai et al., 2025) | – | 29.7 | 41.5 | 11.1 | 16.9 | 19.2 | 3.3 | 36.5 | 22.6 |
| + DCPO (Yang et al., 2025a) | 0.29 | 38.5 | 55.2 | 16.1 | 23.8 | 26.4 | 6.5 | 38.0 | 29.2 |
| + GRPO (Shao et al., 2024) | 0.37 | 36.8 | 52.6 | 15.3 | 22.4 | 24.8 | 5.9 | 37.4 | 27.9 |
| + AVSPO (Ours) | **0.14** | **42.7** | **61.3** | **18.9** | **26.5** | **29.5** | **7.8** | **38.6** | **32.2** |
| Qwen2.5-3B-Instruct | – | 63.0 | 76.5 | 25.7 | 24.4 | 27.4 | 5.0 | 37.7 | 37.1 |
| + INTUITOR (Zhao et al., 2025) | – | 65.3 | 61.4 | 26.9 | 27.6 | 30.8 | 7.6 | 36.9 | 36.6 |
| + RENT (Prabhudesai et al., 2025) | – | 64.1 | 56.2 | 26.1 | 25.8 | 28.9 | 6.4 | 36.4 | 34.8 |
| + DCPO (Yang et al., 2025a) | 0.27 | 70.4 | 71.2 | 29.3 | 32.8 | 35.6 | 10.8 | 37.8 | 41.1 |
| + GRPO (Shao et al., 2024) | 0.35 | 68.9 | 68.7 | 28.4 | 31.2 | 33.7 | 9.8 | 37.2 | 39.7 |
| + AVSPO (Ours) | **0.13** | **73.6** | **75.8** | **31.2** | **35.1** | **37.4** | **12.3** | **38.4** | **43.4** |
| Qwen2.5-14B[Base] | – | 68.4 | 89.2 | 28.5 | 38.2 | 42.1 | 16.8 | 56.7 | 48.6 |
| + INTUITOR (Zhao et al., 2025) | – | 73.2 | 68.9 | 31.4 | 43.5 | 46.3 | 21.2 | 55.8 | 48.6 |
| + RENT (Prabhudesai et al., 2025) | – | 70.8 | 64.5 | 29.6 | 40.8 | 44.1 | 19.4 | 55.2 | 46.3 |
| + DCPO (Yang et al., 2025a) | 0.21 | 75.6 | 74.2 | 34.5 | 47.8 | 50.6 | 25.4 | 56.8 | 52.1 |
| + GRPO (Shao et al., 2024) | 0.28 | 72.5 | 71.8 | 32.1 | 44.6 | 48.2 | 23.8 | 56.2 | 49.9 |
| + AVSPO (Ours) | **0.11** | **78.9** | **77.4** | **36.8** | **50.3** | **53.1** | **27.5** | **57.4** | **54.5** |
| *Math-Specialized Models* | | | | | | | | | |
| Qwen2.5-Math-1.5B[Base] | – | 31.8 | 80.2 | 11.4 | 22.2 | 27.0 | 3.2 | 27.5 | 29.0 |
| + INTUITOR (Zhao et al., 2025) | – | 52.4 | 43.6 | 19.7 | 31.4 | 35.2 | 8.5 | 26.8 | 31.1 |
| + RENT (Prabhudesai et al., 2025) | – | 47.8 | 38.4 | 17.2 | 28.5 | 32.8 | 6.9 | 26.3 | 28.3 |
| + DCPO (Yang et al., 2025a) | 0.31 | 61.4 | 52.6 | 22.8 | 33.5 | 39.2 | 11.8 | 27.6 | 35.6 |
| + GRPO (Shao et al., 2024) | 0.40 | 58.6 | 49.8 | 19.2 | 31.7 | 37.6 | 10.6 | 27.0 | 33.5 |
| + AVSPO (Ours) | **0.15** | **67.2** | **59.3** | **28.9** | **37.8** | **41.6** | **14.2** | **28.2** | **39.6** |
| Qwen2.5-Math-7B[Base] | – | 60.8 | 86.3 | 20.2 | 30.4 | 35.0 | 13.3 | 39.4 | 40.8 |
| + INTUITOR (Zhao et al., 2025) | – | 68.9 | 62.5 | 25.3 | 37.1 | 39.2 | 18.4 | 38.6 | 41.4 |
| + RENT (Prabhudesai et al., 2025) | – | 65.4 | 57.8 | 23.1 | 34.6 | 37.5 | 16.1 | 38.1 | 38.9 |
| + DCPO (Yang et al., 2025a) | 0.25 | 69.8 | 66.4 | 27.2 | 39.5 | 41.2 | 21.5 | 39.5 | 43.6 |
| + GRPO (Shao et al., 2024) | 0.33 | 65.0 | 65.3 | 25.7 | 36.2 | 40.9 | 20.6 | 38.9 | 42.2 |
| + AVSPO (Ours) | **0.14** | **74.1** | **69.7** | **29.4** | **43.6** | **43.8** | **23.2** | **40.1** | **45.9** |

[‡] avg@32 to reduce variance from limited problem counts.

AVSPO's stratified approach achieves both lowest ACR (0.15) and highest accuracy (+8.6%), validating the importance of structured reward diversity.

*Table 2.* Ablation study on virtual sample construction strategies. All experiments use Qwen2.5-Math-1.5B trained for 500 steps. GRPO serves as the baseline.

| Strategy | ACR (↓) | MATH-500 (↑) |
|---|---|---|
| No augmentation (GRPO) | 0.40 | 58.6 |
| Random uniform | 0.22 −0.18 | 62.1 +3.5 |
| Fixed partial ($r_v = 0.5$) | 0.19 −0.21 | 63.5 +4.9 |
| Exponential decay | 0.18 −0.22 | 64.2 +5.6 |
| **Stratified (Ours)** | **0.15** −0.25 | **67.2** +8.6 |

**Mechanism Isolation.** We further separate the two collapsed-group cases targeted by AVSPO. Error-Only augments only all-incorrect groups, while Correct-Only augments only all-correct groups. Table 3 shows that both interventions are independently useful: Error-Only mainly

suppresses all-wrong collapse, Correct-Only mainly suppresses all-correct collapse, and the full method combines both effects for the largest gain.

*Table 3.* Mechanism isolation on Qwen2.5-Math-1.5B trained for 500 steps on MATH-500 over 5 seeds.

| Method | Acc. | All-Wrong | All-Correct |
|---|---|---|---|
| GRPO | 58.6±1.4 | 24.8% | 15.2% |
| Error-Only | 63.2±1.3 | 9.1% | 14.5% |
| Correct-Only | 60.8±1.5 | 23.6% | 4.2% |
| **Full AVSPO** | **67.2±1.2** | **8.7%** | **6.3%** |

**Adaptive vs. Fixed Thresholding.** We compare three fixed thresholds ($\tau \in \{0.3, 0.5, 0.7\}$) against our adaptive mechanism (Table 4). Fixed thresholds face a fundamental trade-off: low values ($\tau = 0.3$) trigger excessive intervention with high variance; high values ($\tau = 0.7$) miss early-stage collapse. The optimal fixed threshold ($\tau = 0.5$) requires

380 steps to reach 60% accuracy, while adaptive thresholding achieves this in 295 steps (22% faster) with 1.8% higher final accuracy. The adaptive mechanism adjusts based on training progress, avoiding both over- and under-intervention.

*Table 4.* Comparison of fixed vs. adaptive thresholding on Qwen2.5-Math-1.5B. "Steps to 60%" denotes training steps to reach 60% accuracy on MATH-500.

| Threshold | Steps to 60% | Final Acc. | Stability |
|---|---|---|---|
| Fixed $\tau = 0.3$ | 420 | 64.8 | High variance |
| Fixed $\tau = 0.5$ | 380 | 65.4 | Moderate |
| Fixed $\tau = 0.7$ | 350 | 63.9 | Under-intervene |
| **Adaptive (Ours)** | **295** | **67.2** | **Stable** |

**Sensitivity Parameter $\alpha$.** The number of virtual samples scales as $K \propto \text{ACR}^\alpha$. We find $\alpha = 0.5$ optimal: smaller values (0.3) under-augment, leaving residual collapse (ACR$= 0.24$); larger values (1.0) over-augment, diluting real gradient signals (accuracy drops 2.1%). The square-root scaling provides appropriate adaptive strength.

**Robustness to Hyperparameters.** AVSPO shows stable performance across parameter ranges. Varying $\tau_{\text{adapt}}^{(0)} \in [0.3, 0.7]$ and $\eta \in [0.005, 0.02]$ yields less than 1% accuracy variation, indicating robustness without extensive tuning.

### 5.5. Filtering versus Augmentation

We finally compare two paradigms for handling collapsed groups: removing them from the update or augmenting their reward statistics. Filtering avoids zero-advantage updates, but the collapsed groups have already consumed rollout computation; dropping them leaves those samples unused, while resampling adds extra rollouts. AVSPO keeps the original generated samples and modifies only the normalization statistics, allowing these groups to provide a learning signal with negligible additional cost.

*Table 5.* Filtering vs. augmentation on Qwen2.5-Math-7B.

| Method | GSM8K | MATH | Util. | Cost |
|---|---|---|---|---|
| GRPO | 65.3±1.8 | 65.0±1.4 | 100% | 1.0× |
| Filter-Drop | 66.2±1.5 | 67.1±1.3 | 62.4% | 1.0× |
| DAPO | 65.8±1.7 | 68.3±1.6 | 54.4% | 1.8× |
| **AVSPO (Ours)** | **69.7±1.4** | **74.1±1.2** | **100%** | **1.0×** |

Table 5 reports a controlled comparison on MATH-Level3-500 with 500 training steps, $G = 8$, $T = 1.0$, and 3 seeds. Filter-Drop improves over GRPO but uses only 62.4% of generated samples. DAPO's dynamic-sampling strategy (Yu et al., 2026) improves MATH accuracy but raises rollout cost to 1.8×. AVSPO obtains the strongest result while retaining 100% sample utilization and 1.0× cost, suggesting that repairing collapsed groups is a more efficient use of generated trajectories than filtering them out.

## 6. Conclusion

We have identified *advantage collapse*, a failure mode in GRPO where homogeneous within-group rewards cause vanishing gradients, as a fundamental bottleneck in reinforcement learning for LLMs. To diagnose this, we introduced the ACR, a zero-overhead metric that quantifies gradient ineffectiveness and predicts final performance. Guided by real-time ACR monitoring, our AVSPO algorithm adaptively injects virtual reward samples to enable learning from collapsed groups, achieving 58–63% reduction in collapse rate and consistent accuracy improvements across 0.5B–14B models on mathematical reasoning benchmarks. We hope ACR becomes a standard diagnostic in binary-reward RLVR pipelines with deterministic verifiers, enabling more transparent and efficient training in this setting.

## Acknowledgments

This work was supported by the National Natural Science Foundation of China under Grant 62306331, 62407037, and CAAI Youth Talent Lifting Project under Grant CAAI2023-2025QNRC001.

## Impact Statement

This work aims to advance the field of RLVR for LLMs by providing diagnostic tools and algorithmic improvements for more stable and efficient training. ACR enables practitioners to identify training inefficiencies in real-time, potentially reducing computational waste and its associated energy footprint, and improving transparency in LLM alignment pipelines. We hope that our contributions will facilitate more efficient development of reasoning capabilities in language models and encourage further research on training diagnostics for policy optimization.

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

# A. Implementation Details

Table 6 summarizes all hyperparameter settings used in our experiments.

*Table 6.* Complete hyperparameter settings for all experiments.

| Hyperparameter | Value |
|---|---|
| *Hardware & Software* | |
| GPUs | 8×NVIDIA A800-80GB |
| Framework | TRL (von Werra et al., 2020) |
| *Sampling* | |
| Group size $G$ | 8 |
| Sampling temperature (training) | 1.0 |
| Sampling temperature (evaluation) | 0 (greedy) |
| Max sequence length | 1,024 tokens |
| *Optimization* | |
| Optimizer | AdamW |
| Learning rate $\eta_\theta$ | $10^{-6}$ |
| $(\beta_1, \beta_2)$ | (0.9, 0.999) |
| Weight decay | 0.01 |
| Gradient clipping norm | 1.0 |
| Training steps | 500 |
| Batch size per GPU | 1 |
| *GRPO/AVSPO* | |
| KL penalty $\beta_{\text{KL}}$ | 0 |
| Clipping range $\varepsilon$ | 0.2 |
| *AVSPO-Specific* | |
| Initial threshold $\tau_{\text{adapt}}^{(0)}$ | 0.5 |
| Sensitivity $\alpha$ | 0.5 |
| Threshold learning rate $\eta$ | 0.01 |
| Collapse threshold $\tau$ | $10^{-6}$ |
| Threshold bounds $[\tau_{\min}, \tau_{\max}]$ | [0.1, 0.9] |
| Anchor reward $r_{\text{anchor}}$ | 0.1 |

**Training Time.** Training a 1.5B model for 500 steps takes approximately 2.5 hours wall-clock time; 7B models require 4 hours; 14B models require approximately 8 hours.

## A.1. Training Data Construction

We construct Level3-500 through difficulty-based selection using Qwen2.5-Math-1.5B as the selector model, inspired by curriculum learning principles (Bengio et al., 2009) and recent findings that overly difficult examples hinder alignment (Hemmat et al., 2025). Specifically, we partition the MATH training split (Hendrycks et al., 2021) into seven difficulty levels (Level 0–6) based on the selector's success rate: Level 0 (95% accuracy), Level 1 (80%), Level 2 (50–80%), Level 3 (30–50%), Level 4 (10–30%), Level 5 (1–10%), and Level 6 (<1%). From Level 3, we randomly sample 500 problems, targeting the optimal learning zone where the model has sufficient but not excessive success rate to generate diverse reward signals. Note that Level3-500 is drawn exclusively from the *training* split, while our evaluation benchmark MATH-500 uses the official *test* split, ensuring no data leakage.

**Training Protocol.** All models are trained for one epoch (500 steps), justified by learning curves plateauing within 400 steps (see Section G). As shown in Figures 6–9, both ACR and accuracy stabilize well before 500 steps across all configurations, confirming that 500 problems provide sufficient training signal for convergence. We report single-run results for the full experimental matrix (6 models × 7 benchmarks × 5 methods) due to computational cost. To validate robustness, we conduct multi-seed experiments ($n = 5$) on four representative models (Qwen2.5-0.5B, Qwen2.5-Math-1.5B, Qwen2.5-3B, and

Qwen2.5-14B) on MATH-500 and GSM8K.

# B. Proofs

*Proof of Proposition 4.2.* By definition, $\hat{A}_i = (r_i - \mu_{\mathcal{R}})/(\sigma_{\mathcal{R}} + \epsilon)$. Thus:

$$\sum_{i=1}^{G} \hat{A}_i^2 = \sum_{i=1}^{G} \frac{(r_i - \mu_{\mathcal{R}})^2}{(\sigma_{\mathcal{R}} + \epsilon)^2} = \frac{G\sigma_{\mathcal{R}}^2}{(\sigma_{\mathcal{R}} + \epsilon)^2}$$

using the definition $\sigma_{\mathcal{R}}^2 = \frac{1}{G} \sum_i (r_i - \mu_{\mathcal{R}})^2$. □

## B.1. Virtual Samples in Homogeneous Groups: Directionality and Stability

This subsection provides a deeper analysis of the bias and stability properties introduced by the virtual reward samples in Equations (9) and (12), focusing on the two degenerate (collapsed) cases where all rollouts in a group are either incorrect or correct.

**Setup.** Fix a prompt $q$ (corresponding to some group $j$). Let $\mathcal{Y}(q)$ denote the space of all trajectories (responses) that the policy can generate. Define the *success* and *failure* subsets

$$\mathcal{S}_q := \{y \in \mathcal{Y}(q) : r(q, y) = 1\}, \qquad \mathcal{F}_q := \{y \in \mathcal{Y}(q) : r(q, y) = 0\}. \tag{17}$$

We denote the success probability by

$$p_\theta(q) := \pi_\theta(\mathcal{S}_q \mid q), \qquad 1 - p_\theta(q) = \pi_\theta(\mathcal{F}_q \mid q). \tag{18}$$

**Notational convention.** Throughout this appendix, we fix a single group $j$ and suppress the group index when the context is unambiguous, writing $\mathcal{R}$, $\sigma_{\mathcal{R}}$, $\mu_{\mathcal{R}}$ instead of $\mathcal{R}_j$, $\sigma_{\mathcal{R}_j}$, $\mu_{\mathcal{R}_j}$. When augmented reward sets are involved, we write $\mathcal{R}'$, $\mu_{\mathcal{R}'}$, $\sigma_{\mathcal{R}'}$ for the quantities after virtual sample injection. The subscript $j$ is retained where explicitly needed for clarity.

**Lemma B.1** (Conditional score identity). *Let $\mathcal{E} \subseteq \mathcal{Y}(q)$ be any measurable set such that $\pi_\theta(\mathcal{E} \mid q) > 0$. Then,*

$$\mathbb{E}_{y \sim \pi_\theta(\cdot \mid q,\, y \in \mathcal{E})}\big[\nabla_\theta \log \pi_\theta(y \mid q)\big] = \nabla_\theta \log \pi_\theta(\mathcal{E} \mid q). \tag{19}$$

*Proof.* Let $Z_\theta := \pi_\theta(\mathcal{E} \mid q) = \sum_{y \in \mathcal{E}} \pi_\theta(y \mid q)$. Using $\nabla_\theta \log \pi_\theta(y \mid q) = \nabla_\theta \pi_\theta(y \mid q)/\pi_\theta(y \mid q)$, we have

$$\mathbb{E}_{y \sim \pi_\theta(\cdot \mid q,\, y \in \mathcal{E})}\big[\nabla_\theta \log \pi_\theta(y \mid q)\big] = \frac{1}{Z_\theta} \sum_{y \in \mathcal{E}} \pi_\theta(y \mid q)\, \nabla_\theta \log \pi_\theta(y \mid q)$$

$$= \frac{1}{Z_\theta} \sum_{y \in \mathcal{E}} \nabla_\theta \pi_\theta(y \mid q) = \frac{\nabla_\theta Z_\theta}{Z_\theta} = \nabla_\theta \log Z_\theta,$$

which proves Equation (19). □

**Lemma B.2** (Uniform advantages for collapsed binary-reward groups). *Assume binary rewards $r(q, y) \in \{0, 1\}$ and consider a* collapsed *group of size $G$ where $r(q, y^{(i)}) = c$ for all $i \in \{1, \ldots, G\}$ and some $c \in \{0, 1\}$. Under AVSPO, the recomputed advantages in Equation (12) satisfy*

$$\hat{A}_j^{(i)\prime} \equiv \hat{A}_c^{(K)} := \frac{c - \mu_{\mathcal{R}_j'}}{\sigma_{\mathcal{R}_j'} + \epsilon}, \qquad \forall i \in \{1, \ldots, G\}, \tag{20}$$

*i.e., all $G$ real samples share a* common *advantage sign in the collapsed cases.*

*Moreover, for the stratified virtual rewards in Equation (9) (with $K \geq 1$) the augmented means are*

$$\mu_{\mathcal{R}_j'} = \begin{cases} \dfrac{G + \frac{K}{2}}{G + K}, & c = 1, \\[2ex] \dfrac{r_{anchor}(K + 1)}{2(G + K)}, & c = 0, \end{cases} \tag{21}$$

*and the augmented standard deviations admit the lower bounds*

$$\sigma_{\mathcal{R}'_j} \geq \begin{cases} \dfrac{\sqrt{GK}}{2(G+K)}, & c = 1, \\ \dfrac{r_{anchor}(K+1)}{2(G+K)}\sqrt{\dfrac{G}{K}}, & c = 0. \end{cases} \tag{22}$$

*Proof.* In a collapsed binary-reward group, all observed rewards equal $c$, hence $\{r_j^{(i)}\}_{i=1}^G$ are identical. Therefore, in Equation (12) each numerator $r_j^{(i)} - \mu_{\mathcal{R}'_j}$ is identical, which proves the constancy claim Equation (20).

We next compute $\mu_{\mathcal{R}'_j}$ in the two collapsed cases. If $c = 1$, then $r_{obs} = \max(\mathcal{R}_j) = 1$ and Equation (9) gives $r_{v_k} = 1 - \frac{k}{K+1} = \frac{K+1-k}{K+1}$. Thus $\sum_{k=1}^K r_{v_k} = \sum_{m=1}^K \frac{m}{K+1} = \frac{K}{2}$, yielding $\mu_{\mathcal{R}'_j} = \frac{G \cdot 1 + \frac{K}{2}}{G+K}$. If $c = 0$, then $r_{obs} = 0$ and Equation (9) gives $r_{v_k} = r_{anchor}\frac{K-k+1}{K}$. Hence $\sum_{k=1}^K r_{v_k} = \frac{r_{anchor}}{K}\sum_{m=1}^K m = \frac{r_{anchor}(K+1)}{2}$, and $\mu_{\mathcal{R}'_j} = \frac{\frac{r_{anchor}(K+1)}{2}}{G+K}$. This proves Equation (21).

For the lower bounds on $\sigma_{\mathcal{R}'_j}$, we use that the total variance is at least the *between-component* variance of a two-component mixture. In the $c = 1$ case, all $G$ real rewards equal 1 and the $K$ virtual rewards have mean $1/2$, so the between-component variance is $\frac{G}{G+K} \cdot \frac{K}{G+K} \cdot (1 - \frac{1}{2})^2 = \frac{GK}{4(G+K)^2}$, implying $\sigma_{\mathcal{R}'_j} \geq \frac{\sqrt{GK}}{2(G+K)}$. In the $c = 0$ case, the real rewards equal 0 and the virtual rewards have mean $\mu_v = \frac{1}{K}\sum_{k=1}^K r_{v_k} = \frac{r_{anchor}(K+1)}{2K}$, so the between-component variance is $\frac{G}{G+K} \cdot \frac{K}{G+K} \cdot \mu_v^2 = \frac{GK}{(G+K)^2} \cdot \frac{r_{anchor}^2(K+1)^2}{4K^2}$, implying $\sigma_{\mathcal{R}'_j} \geq \frac{r_{anchor}(K+1)}{2(G+K)}\sqrt{\frac{G}{K}}$. This proves Equation (22). $\qquad\square$

**Proposition B.3** (Bounded magnitude of uniform updates). *Under the assumptions of Theorem B.2 and $1 \leq K \leq G$, the common advantage magnitude is bounded by*

$$\left|\hat{A}_c^{(K)}\right| \leq \sqrt{\frac{K}{G}} \leq 1. \tag{23}$$

*Proof.* When $c = 1$, Equation (21) gives $1 - \mu_{\mathcal{R}'_j} = \frac{K}{2(G+K)}$ and Equation (22) gives $\sigma_{\mathcal{R}'_j} \geq \frac{\sqrt{GK}}{2(G+K)}$. Thus,

$$\left|\hat{A}_1^{(K)}\right| = \frac{1 - \mu_{\mathcal{R}'_j}}{\sigma_{\mathcal{R}'_j} + \epsilon} \leq \frac{1 - \mu_{\mathcal{R}'_j}}{\sigma_{\mathcal{R}'_j}} \leq \frac{\frac{K}{2(G+K)}}{\frac{\sqrt{GK}}{2(G+K)}} = \sqrt{\frac{K}{G}}.$$

When $c = 0$, Equation (21) gives $\mu_{\mathcal{R}'_j} = \frac{r_{anchor}(K+1)}{2(G+K)}$ and Equation (22) gives $\sigma_{\mathcal{R}'_j} \geq \frac{r_{anchor}(K+1)}{2(G+K)}\sqrt{\frac{G}{K}}$. Therefore,

$$\left|\hat{A}_0^{(K)}\right| = \frac{\mu_{\mathcal{R}'_j}}{\sigma_{\mathcal{R}'_j} + \epsilon} \leq \frac{\mu_{\mathcal{R}'_j}}{\sigma_{\mathcal{R}'_j}} \leq \frac{\frac{r_{anchor}(K+1)}{2(G+K)}}{\frac{r_{anchor}(K+1)}{2(G+K)}\sqrt{\frac{G}{K}}} = \sqrt{\frac{K}{G}}.$$

Combining the two cases proves Equation (23). $\qquad\square$

**Theorem B.4** (Principled direction of AVSPO updates in homogeneous groups). *Fix a prompt $q$ and consider the (unclipped, on-policy) per-prompt gradient contribution*

$$g_c(q) := \frac{1}{G}\sum_{i=1}^G \hat{A}_c^{(K)} \nabla_\theta \log \pi_\theta(y^{(i)} \mid q), \tag{24}$$

*where $\hat{A}_c^{(K)}$ is the common advantage from Equation (20). Then, conditioned on an all-incorrect group ($c = 0$) or an all-correct group ($c = 1$), we have*

$$\mathbb{E}\left[g_0(q) \,\middle|\, y^{(i)} \in \mathcal{F}_q \,\forall i\right] = \hat{A}_0^{(K)} \nabla_\theta \log\left(1 - p_\theta(q)\right), \tag{25}$$

$$\mathbb{E}\left[g_1(q) \,\middle|\, y^{(i)} \in \mathcal{S}_q \,\forall i\right] = \hat{A}_1^{(K)} \nabla_\theta \log p_\theta(q). \tag{26}$$

*In particular, $\hat{A}_0^{(K)} < 0$ and $\hat{A}_1^{(K)} > 0$, so gradient ascent decreases $\log(1 - p_\theta(q))$ in the all-incorrect case and increases $\log p_\theta(q)$ in the all-correct case.*

*Proof.* We first note the signs: when $c = 0$, Equation (21) gives $\mu_{\mathcal{R}'_j} > 0$ hence $\hat{A}_0^{(K)} = (0 - \mu_{\mathcal{R}'_j})/(\sigma_{\mathcal{R}'_j} + \epsilon) < 0$; when $c = 1$, Equation (21) gives $\mu_{\mathcal{R}'_j} < 1$ hence $\hat{A}_1^{(K)} > 0$.

For Equation (25), under the conditioning $y^{(i)} \in \mathcal{F}_q$ for all $i$, each $y^{(i)}$ is distributed as $\pi_\theta(\cdot \mid q, y \in \mathcal{F}_q)$ and remains i.i.d. Therefore,

$$\mathbb{E}\Big[g_0(q) \,\Big|\, y^{(i)} \in \mathcal{F}_q \,\forall i\Big] = \hat{A}_0^{(K)} \, \mathbb{E}_{y \sim \pi_\theta(\cdot \mid q, \, y \in \mathcal{F}_q)}\big[\nabla_\theta \log \pi_\theta(y \mid q)\big].$$

Applying the conditional score identity Theorem B.1 with $\mathcal{E} = \mathcal{F}_q$ yields

$$\mathbb{E}_{y \sim \pi_\theta(\cdot \mid q, \, y \in \mathcal{F}_q)}\big[\nabla_\theta \log \pi_\theta(y \mid q)\big] = \nabla_\theta \log \pi_\theta(\mathcal{F}_q \mid q) = \nabla_\theta \log\big(1 - p_\theta(q)\big),$$

which proves Equation (25). The proof of Equation (26) is identical with $\mathcal{E} = \mathcal{S}_q$, noting $\pi_\theta(\mathcal{S}_q \mid q) = p_\theta(q)$. $\qquad\square$

**Lemma B.5** (Clipping-induced gradient gating). *Consider the per-token PPO/GRPO term (cf. Equations (3) and (13))*

$$\ell(\rho; A) := \min\Big(\rho A, \; \mathrm{clip}(\rho, 1 - \varepsilon, 1 + \varepsilon)\, A\Big), \tag{27}$$

*where $\rho$ is a policy ratio and $A$ is an advantage (here $A = \hat{A}'_i$). Then:*

$$A > 0, \; \rho \geq 1 + \varepsilon \quad \implies \quad \ell(\rho; A) = (1 + \varepsilon)A \text{ and } \nabla_\theta \ell(\rho; A) = 0, \tag{28}$$
$$A < 0, \; \rho \leq 1 - \varepsilon \quad \implies \quad \ell(\rho; A) = (1 - \varepsilon)A \text{ and } \nabla_\theta \ell(\rho; A) = 0. \tag{29}$$

*Proof.* We prove Equation (28); Equation (29) follows analogously. When $A > 0$ and $\rho \geq 1 + \varepsilon$, we have $\mathrm{clip}(\rho, 1 - \varepsilon, 1 + \varepsilon) = 1 + \varepsilon \leq \rho$. Multiplying by $A > 0$ preserves the inequality, so $(1 + \varepsilon)A \leq \rho A$ and hence $\ell(\rho; A) = \min(\rho A, (1 + \varepsilon)A) = (1 + \varepsilon)A$, which is constant w.r.t. $\theta$. Therefore its gradient vanishes, i.e., $\nabla_\theta \ell(\rho; A) = 0$. $\qquad\square$

*Remark* B.6 (Implications for the reviewer concern: collapse vs. sharpening). In the all-incorrect case, Theorem B.4 shows that the *expected* AVSPO update is aligned with decreasing $\log(1 - p_\theta(q))$, i.e., reducing the probability mass assigned to the failure set $\mathcal{F}_q$ rather than "collapsing" probability without direction. In the all-correct case, the update increases $\log p_\theta(q)$ but its magnitude is bounded (Theorem B.3), and the PPO clipping term becomes flat once token-level ratios exceed $1 + \varepsilon$ (Theorem B.5), preventing unbounded over-sharpening from a single group.

## B.2. Bias Upper Bound and Convergence Implication

**Assumption B.7** (Bounded score function). There exists a constant $B > 0$ such that for all prompts $q$ and trajectories $y \in \mathcal{Y}(q)$,

$$\big\|\nabla_\theta \log \pi_\theta(y \mid q)\big\| \leq B. \tag{30}$$

**Proposition B.8** (Bias upper bound of AVSPO relative to GRPO). *Consider one training iteration $n$ with a batch of $N$ groups and the corresponding $ACR^{(n)}$ computed by Equation (6). Let $K^{(n)}$ be the number of virtual samples used by AVSPO in Equation (8). Define the (unclipped, on-policy) per-batch gradient estimators*

$$\hat{g}^{GRPO}(\theta) := \frac{1}{N} \sum_{j=1}^{N} \frac{1}{G} \sum_{i=1}^{G} \hat{A}_j^{(i)} \nabla_\theta \log \pi_\theta(y_j^{(i)} \mid q_j), \tag{31}$$

$$\hat{g}^{AVSPO}(\theta) := \frac{1}{N} \sum_{j=1}^{N} \frac{1}{G} \sum_{i=1}^{G} \hat{A}_j^{(i)\prime} \nabla_\theta \log \pi_\theta(y_j^{(i)} \mid q_j), \tag{32}$$

*where $\hat{A}_j^{(i)}$ and $\hat{A}_j^{(i)\prime}$ are computed by Equations (2) and (12).*

*Under binary rewards and Theorem B.7, the per-iteration gradient discrepancy satisfies:*

$$\big\|\hat{g}^{AVSPO}(\theta) - \hat{g}^{GRPO}(\theta)\big\| \leq B \sqrt{\frac{K^{(n)}}{G}} \cdot ACR^{(n)} \leq B \cdot ACR^{(n)}. \tag{33}$$

*Consequently, the conditional (one-step) bias magnitude is bounded by*

$$\left\| \mathbb{E}\left[\hat{g}^{AVSPO}(\theta) - \hat{g}^{GRPO}(\theta) \mid \theta\right] \right\| \leq B\, \mathbb{E}\left[ \sqrt{\frac{K^{(n)}}{G}}\, ACR^{(n)} \,\middle|\, \theta \right]. \tag{34}$$

*Proof.* If augmentation is not triggered at iteration $n$, then $\hat{A}_j^{(i)\prime} = \hat{A}_j^{(i)}$ for all groups, hence $\hat{g}^{AVSPO} = \hat{g}^{GRPO}$ and the bound is trivial.

Assume augmentation is triggered. Then AVSPO modifies advantages only for groups with $\sigma_{\mathcal{R}_j} < \tau$. Let $\mathcal{C}^{(n)} := \{j : \sigma_{\mathcal{R}_j} < \tau\}$ denote the set of collapsed groups in this batch, so $|\mathcal{C}^{(n)}|/N = ACR^{(n)}$ by Equation (6). For any $j \in \mathcal{C}^{(n)}$, GRPO yields $\hat{A}_j^{(i)} = 0$ for all $i$ (cf. Equation (4)), while AVSPO yields a common advantage $\hat{A}_j^{(i)\prime} \equiv \hat{A}_c^{(K^{(n)})}$ (cf. Theorem B.2). Therefore,

$$\hat{g}^{AVSPO}(\theta) - \hat{g}^{GRPO}(\theta) = \frac{1}{N}\sum_{j \in \mathcal{C}^{(n)}} \frac{1}{G}\sum_{i=1}^{G} \hat{A}_c^{(K^{(n)})}\, \nabla_\theta \log \pi_\theta(y_j^{(i)} \mid q_j).$$

Taking norms and applying Equation (30) gives

$$\left\| \hat{g}^{AVSPO}(\theta) - \hat{g}^{GRPO}(\theta) \right\| \leq \frac{1}{N}\sum_{j \in \mathcal{C}^{(n)}} \frac{1}{G}\sum_{i=1}^{G} \left| \hat{A}_c^{(K^{(n)})} \right|\, \left\| \nabla_\theta \log \pi_\theta(y_j^{(i)} \mid q_j) \right\|$$

$$\leq \frac{|\mathcal{C}^{(n)}|}{N}\, \left| \hat{A}_c^{(K^{(n)})} \right|\, B = ACR^{(n)}\, \left| \hat{A}_c^{(K^{(n)})} \right|\, B.$$

Finally, Theorem B.3 gives $\left| \hat{A}_c^{(K^{(n)})} \right| \leq \sqrt{K^{(n)}/G} \leq 1$ (since $K^{(n)} \leq G$), which yields Equation (33). The conditional bound Equation (34) follows by taking conditional expectation on both sides. $\square$

**Assumption B.9** (Smoothness and boundedness for the reference objective)**.** Let $F(\theta)$ denote the (population) objective whose stationary points are regarded as the "correct" fixed points for the baseline algorithm (e.g., $F = \mathcal{J}^{GRPO}$ under the standard PPO/GRPO analysis). Assume $F$ is $L$-smooth and upper-bounded:

$$\|\nabla F(\theta) - \nabla F(\theta')\| \leq L\|\theta - \theta'\|, \qquad F(\theta) \leq F_{\max} < \infty. \tag{35}$$

**Theorem B.10** (Convergence to an approximate stationary point under bounded bias)**.** *Consider the stochastic ascent update*

$$\theta^{(n+1)} = \theta^{(n)} + \eta_\theta\, \hat{g}^{(n)}, \tag{36}$$

*where $\hat{g}^{(n)}$ is the stochastic gradient used by AVSPO at iteration $n$. Assume Theorem B.9 and that for each $n$,*

$$\mathbb{E}\left[\hat{g}^{(n)} \mid \theta^{(n)}\right] = \nabla F(\theta^{(n)}) + b^{(n)}, \qquad \mathbb{E}\left[ \left\| \hat{g}^{(n)} - \mathbb{E}[\hat{g}^{(n)} \mid \theta^{(n)}] \right\|^2 \,\middle|\, \theta^{(n)} \right] \leq \sigma^2, \tag{37}$$

*for some (possibly iteration-dependent) bias vector $b^{(n)}$ and noise level $\sigma^2$. If $\eta_\theta \leq \frac{1}{4L}$, then for any $T \geq 1$,*

$$\frac{1}{T}\sum_{n=0}^{T-1} \mathbb{E}\left[ \left\| \nabla F(\theta^{(n)}) \right\|^2 \right] \leq \frac{4\left(F_{\max} - F(\theta^{(0)})\right)}{\eta_\theta T} + 2L\eta_\theta \sigma^2 + \frac{3}{T}\sum_{n=0}^{T-1} \mathbb{E}\left[ \|b^{(n)}\|^2 \right]. \tag{38}$$

*In particular, if $\|b^{(n)}\| \leq b$ uniformly, then the iterates converge to an $O(b)$-stationary region; and if $\|b^{(n)}\| \to 0$, then $\liminf_{n\to\infty} \mathbb{E}\|\nabla F(\theta^{(n)})\| = 0$.*

*Proof.* By $L$-smoothness of $F$, for $\Delta^{(n)} := \theta^{(n+1)} - \theta^{(n)} = \eta_\theta \hat{g}^{(n)}$ we have

$$F(\theta^{(n+1)}) \geq F(\theta^{(n)}) + \left\langle \nabla F(\theta^{(n)}), \Delta^{(n)} \right\rangle - \frac{L}{2}\|\Delta^{(n)}\|^2.$$

Taking conditional expectation given $\theta^{(n)}$ and using Equation (37) yields

$$\mathbb{E}\left[F(\theta^{(n+1)}) \mid \theta^{(n)}\right] \geq F(\theta^{(n)}) + \eta_\theta \left\langle \nabla F(\theta^{(n)}), \nabla F(\theta^{(n)}) + b^{(n)} \right\rangle - \frac{L\eta_\theta^2}{2} \mathbb{E}\left[\|\hat{g}^{(n)}\|^2 \mid \theta^{(n)}\right]$$

$$\geq F(\theta^{(n)}) + \eta_\theta \left(\frac{1}{2}\|\nabla F(\theta^{(n)})\|^2 - \frac{1}{2}\|b^{(n)}\|^2\right) - \frac{L\eta_\theta^2}{2}\left(\|\nabla F(\theta^{(n)}) + b^{(n)}\|^2 + \sigma^2\right)$$

$$\geq F(\theta^{(n)}) + \left(\frac{\eta_\theta}{2} - L\eta_\theta^2\right)\|\nabla F(\theta^{(n)})\|^2 - \left(\frac{\eta_\theta}{2} + L\eta_\theta^2\right)\|b^{(n)}\|^2 - \frac{L\eta_\theta^2}{2}\sigma^2,$$

where we used $\langle x, y \rangle \geq -\frac{1}{2}\|x\|^2 - \frac{1}{2}\|y\|^2$ and $\|x + y\|^2 \leq 2\|x\|^2 + 2\|y\|^2$. Under $\eta_\theta \leq \frac{1}{4L}$, we have $\frac{\eta_\theta}{2} - L\eta_\theta^2 \geq \frac{\eta_\theta}{4}$ and $\frac{\eta_\theta}{2} + L\eta_\theta^2 \leq \frac{3\eta_\theta}{4}$, hence

$$\mathbb{E}[F(\theta^{(n+1)})] \geq \mathbb{E}[F(\theta^{(n)})] + \frac{\eta_\theta}{4}\mathbb{E}\|\nabla F(\theta^{(n)})\|^2 - \frac{3\eta_\theta}{4}\mathbb{E}\|b^{(n)}\|^2 - \frac{L\eta_\theta^2}{2}\sigma^2.$$

Summing over $n = 0, \ldots, T - 1$ and using $F(\theta^{(T)}) \leq F_{\max}$ gives Equation (38). □

**Corollary B.11** (Implication for AVSPO: convergence when collapse vanishes). *Let $F = \mathcal{J}^{GRPO}$ in Theorem B.10. If the virtual augmentation becomes rare in the sense that $\mathbb{E}[ACR^{(n)}] \to 0$, then by Theorem B.8 the bias term satisfies $\|b^{(n)}\| \to 0$ (in expectation), and AVSPO converges to the same first-order stationary set as GRPO under the standard smoothness/noise assumptions.*

*Remark* B.12 (Clipping only reduces the discrepancy). All bounds above are derived for the unclipped, on-policy form (cf. Equation (24)). For the actual PPO/GRPO clipped objective, the gradient is further *gated* by clipping (Theorem B.5), which can only reduce the magnitude of per-sample contributions. Therefore, Equation (33) is a conservative upper bound for the implemented AVSPO update.

# C. Algorithm and Computational Complexity

Algorithm 1 presents the complete AVSPO training procedure.

**Computational Complexity.** AVSPO's per-iteration complexity is $O(NG)$, identical to vanilla GRPO. ACR computation requires $O(NG)$ operations for reward variance calculation. Virtual sample generation adds at most $O(NG)$ operations when all groups are collapsed. The memory overhead is negligible: storing $K \leq G$ scalar rewards per collapsed group requires only $O(NG)$ additional floats. No additional LLM forward passes are required, preserving GRPO's ~35% memory advantage over actor-critic methods.

## C.1. Empirical Computational Overhead

To validate the theoretical complexity analysis, we profile the per-step training time breakdown for Qwen2.5-Math-1.5B on Level3-500. Figure 4 presents the results. Table 7 summarizes the detailed timing breakdown.

*Table 7.* Per-step training time breakdown for Qwen2.5-Math-1.5B. Components execute in parallel across 8 GPUs; percentages show relative computational cost. AVSPO's additional overhead is negligible.

| Component | Time (s) | Percentage |
|---|---|---|
| LLM Generation | 76.55 | 48.2% |
| Input Preparation | 77.10 | 48.6% |
| Loss Computation | 2.47 | 1.6% |
| Log Probs & Entropy | 2.18 | 1.4% |
| Reward Calculation | 0.42 | 0.3% |
| **ACR + Virtual Sample** | **0.003** | **<0.01%** |
| **Component Total** | 158.72 | 100% |
| **Wall-clock per Step** | 27.97 | (parallel) |
| **Wall-clock (500 steps)** | 13,987s (3.9h) | — |

**Key Findings:**

**Algorithm 1** AVSPO Training Algorithm

**Require:** Initial policy $\pi_{\theta_0}$, question set $\mathcal{Q}$, group size $G$, sensitivity $\alpha$, initial threshold $\tau_{\text{adapt}}^{(0)}$, threshold learning rate $\eta$, policy learning rate $\eta_\theta$

**Ensure:** Optimized policy $\pi_\theta$

1: **for** iteration $n = 1, 2, \ldots, N_{\text{iter}}$ **do**
2:     *// Stage 1: Sampling & ACR Computation*
3:     Sample batch $\{q_1, \ldots, q_N\} \sim \mathcal{Q}$
4:     **for** each question $q_j$ in batch **do**
5:         Sample $G$ solutions: $\mathcal{O}_j = \{y_j^{(1)}, \ldots, y_j^{(G)}\} \sim \pi_{\theta_{\text{old}}}(\cdot|q_j)$
6:         Evaluate rewards: $\mathcal{R}_j = \{r(q_j, y_j^{(1)}), \ldots, r(q_j, y_j^{(G)})\}$
7:     **end for**
8:     Compute $\text{ACR}^{(n)}$ via Equation 6
9:     *// Stage 2: Virtual Sample Augmentation*
10:     **for** each group $j = 1, \ldots, N$ **do**
11:         **if** $\text{ACR}^{(n)} > \tau_{\text{adapt}}^{(n)}$ and $\sigma_{\mathcal{R}_j} < \tau$ **then**
12:             $K \leftarrow \max(1, \min(G, \lceil G \cdot (\text{ACR}^{(n)})^\alpha \rceil))$ {Eq. 8}
13:             Generate $\mathcal{V}_j = \{v_1, \ldots, v_K\}$ via Equation 9
14:             $\mathcal{R}_j' \leftarrow \mathcal{R}_j \cup \mathcal{V}_j$
15:         **else**
16:             $\mathcal{R}_j' \leftarrow \mathcal{R}_j$
17:         **end if**
18:         Compute advantages $\{\hat{A}_i'\}_{i=1}^G$ via Equation 12 {Only $G$ real samples}
19:     **end for**
20:     *// Stage 3: Parameter Update*
21:     Update policy: $\theta \leftarrow \theta + \eta_\theta \nabla_\theta \mathcal{J}^{\text{AVSPO}}(\theta)$ {Virtual samples do not contribute gradients}
22:     Update threshold: $\tau_{\text{adapt}}^{(n+1)} \leftarrow \tau_{\text{adapt}}^{(n)} + \eta \cdot \text{sign}(\Delta J^{(n)}) \cdot (\text{ACR}^{(n)} - \tau_{\text{adapt}}^{(n)})$
23: **end for**
24: **return** $\pi_\theta$

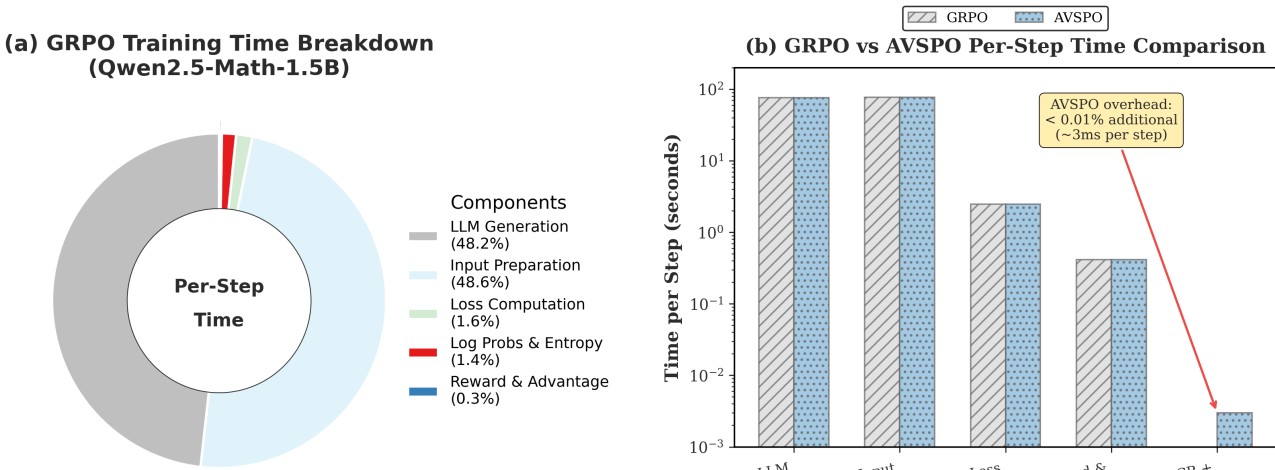

*Figure 4.* **Computational overhead analysis.** (a) Per-step time breakdown for GRPO training on Qwen2.5-Math-1.5B. LLM generation and input preparation dominate ($>96\%$ of total time), while reward and advantage computation account for only 0.3%. (b) Comparison between GRPO and AVSPO. AVSPO adds $<0.01\%$ overhead ($\sim$3ms per step) for ACR monitoring and virtual sample generation, which is negligible compared to the 27.97s per-step total time.

- **LLM-dominated computation**: LLM generation (48.2%) and input preparation (48.6%) account for over 96% of

computational cost. Reward and advantage computation account for only 0.3%. Due to parallel execution across 8 GPUs, wall-clock time per step is 27.97s.

- **Negligible AVSPO overhead**: ACR monitoring and virtual sample generation add approximately 3ms per step ($<$0.01% overhead). For a complete 500-step training run, this translates to only $\sim$1.5 seconds of additional wall-clock time.

- **No additional GPU memory**: AVSPO stores at most $K \leq G$ scalar values per collapsed group, requiring negligible memory compared to model parameters and activations.

- **No additional LLM calls**: Unlike methods that require additional sampling or model queries, AVSPO operates purely on reward statistics already computed during standard GRPO training.

These results confirm that AVSPO achieves its 58–63% reduction in advantage collapse with virtually zero computational cost, making it a practical drop-in replacement for vanilla GRPO.

## D. Prompt Template and Reward Function

Figure 5 illustrates the prompt template employed for all training and evaluation experiments.

---

**Example: Prompt for Math Problems**

<|im_start|>system
You are a helpful assistant that solves MATH problems. Please reason step by step, and put your final answer within \boxed{}.<|im_end|>
<|im_start|>user
Let $a$ and $b$ be nonzero real numbers such that \\[(2 - 7i)(a + bi)\\] is pure imaginary. Find $\\frac{a}{b}.$<|im_end|>
<|im_start|>assistant

---

*Figure 5.* Prompt template for mathematical reasoning tasks. The model is instructed to solve problems step by step and present the final answer within `\boxed{}`.

For base model evaluation, we report the best accuracy obtained using either the prompt template above or the default Qwen (Yang et al., 2024) system prompt: *"Please reason step by step, and put your final answer within \boxed{}."*

**Reward Function.** We employ a binary box accuracy reward:

$$r(q, y) = \mathbb{I}\left[\texttt{extract}(y) = a^*\right] \tag{39}$$

where $\texttt{extract}(\cdot)$ parses the final answer enclosed in `\boxed{}` from the model's response, and $a^*$ denotes the ground-truth answer. The comparison uses symbolic equivalence checking to handle mathematically equivalent expressions (*e.g.*, $\frac{1}{2} = 0.5$). This sparse, outcome-only reward signal is characteristic of the RLVR paradigm and directly contributes to the advantage collapse phenomenon we study.

## E. Code Generation with Deterministic Verifiers

To test whether advantage collapse is specific to mathematical reasoning, we evaluate AVSPO on code generation, another binary-reward RLVR setting with deterministic verification. We train Qwen2.5-Coder-7B on MBPP train-500 and evaluate pass@1 on the MBPP test set over 3 seeds.

*Table 8.* Code generation results on MBPP with test-based deterministic verification.

| Model | Base | GRPO | AVSPO | ACR |
|---|---|---|---|---|
| Qwen2.5-Coder-7B | 76.9 | 83.7$\pm$1.2 | **86.7$\pm$0.9** | 41.2%$\rightarrow$18.5% |

AVSPO improves pass@1 by 3.0 points over GRPO while reducing ACR from 41.2% to 18.5%. This supports applicability beyond math when the task has the same core structure: binary rewards produced by a deterministic verifier. Extending AVSPO to soft rewards or noisy human feedback remains future work.

## F. Sensitivity Analysis Details

This appendix provides detailed analysis of how ACR responds to four key factors that influence reward diversity. For each factor, we vary one hyperparameter while fixing others, measuring $\text{ACR}_{100}$ (mean ACR over the first 100 steps) and final accuracy.

**Model Scale.** Figure 3(a) reveals a general trend where larger models tend to achieve lower ACR and higher accuracy, though the relationship is not strictly monotonic. Smaller models (0.5B, 1.5B) consistently suffer from high ACR ($>0.70$) with accuracy below 35%, while mid-to-large models (3B–14B) achieve substantially lower ACR ($<0.40$) with accuracy above 50%. Notably, the 7B model achieves the lowest ACR (0.15), even lower than the 14B model (0.36), suggesting that the optimal capacity for a given task difficulty may not always be the largest. This pattern reflects that increased capacity generally enables more diverse response generation, but task-model matching also plays a role: when model capability significantly exceeds task difficulty, the model may consistently succeed (all-correct), paradoxically increasing ACR.

**Sampling Temperature.** Figure 3(b) shows that ACR decreases monotonically with temperature, from 0.46 at $T = 0.1$ to 0.08 at $T = 1.0$. However, accuracy exhibits a non-monotonic pattern, peaking at moderate temperatures ($T = 0.3$–$0.5$). This decoupling reveals a key insight: *low ACR is necessary but not sufficient for good performance*. At low temperatures, deterministic sampling produces homogeneous outputs (high ACR); at high temperatures, increased stochasticity ensures reward diversity (low ACR) but introduces excessive randomness that degrades solution quality. The optimal regime balances exploration (diverse outputs, low ACR) with exploitation (coherent reasoning, high accuracy).

**Group Size.** Figure 3(c) demonstrates that increasing $G$ from 2 to 8 reduces ACR from 0.52 to 0.20, with accuracy improving correspondingly. Larger groups increase the probability of observing mixed outcomes: for a problem with success rate $p$, the probability of reward homogeneity (all-correct or all-incorrect) is $p^G + (1 - p)^G$, which decreases exponentially with $G$. However, the gains diminish beyond $G = 6$, while computational cost scales linearly. This suggests $G \in [6, 8]$ as the practical sweet spot, balancing gradient effectiveness against resource constraints.

**Problem Difficulty.** Figure 3(d) reveals a U-shaped relationship: both easy (Level 0–1) and hard (Level 5–6) problems cause high ACR ($>0.45$), while intermediate difficulty (Level 3–4) yields optimal conditions (ACR $\approx 0.30$, accuracy $>55\%$). The mechanism is symmetric: easy problems lead to uniform success ($r_i = 1 \ \forall i$), hard problems lead to uniform failure ($r_i = 0 \ \forall i$), and both result in $\sigma_{\mathcal{R}} = 0$. This finding aligns with curriculum learning principles (Bengio et al., 2009) and validates our choice of Level3-500 for training data construction.

## G. ACR Training Dynamics

This appendix provides detailed visualizations of ACR evolution throughout training under different hyperparameter configurations. These curves complement the summary statistics presented in the main text (Figure 3) by showing the full training dynamics.

### G.1. ACR Dynamics Across Model Scales

Figure 6 shows how ACR evolves during training for models ranging from 0.5B to 14B parameters. Several patterns emerge:

- **Scale-dependent convergence**: Larger models (7B, 14B) rapidly converge to lower ACR ($< 0.35$) within the first 100 steps, while smaller models (0.5B, 1.5B) exhibit persistent high ACR throughout training.

- **Stability**: Larger models show more stable ACR trajectories with less variance, suggesting more consistent reward diversity across batches.

- **Early prediction**: The ACR gap between model scales is apparent within the first 50 steps, enabling early detection of problematic configurations.

## G.2. ACR Dynamics Across Sampling Temperatures

Figure 7 illustrates the effect of sampling temperature on ACR dynamics. Key observations:

- **Temperature-ACR relationship**: Lower temperatures ($T = 0.1, 0.3$) produce deterministic outputs leading to high ACR, while higher temperatures ($T = 0.9, 1.0$) maintain low ACR through increased stochasticity.

- **Accuracy-temperature trade-off**: Despite low ACR at high temperatures, accuracy does not monotonically increase, suggesting an optimal temperature range ($T \approx 0.3$–$0.5$) that balances exploration and exploitation.

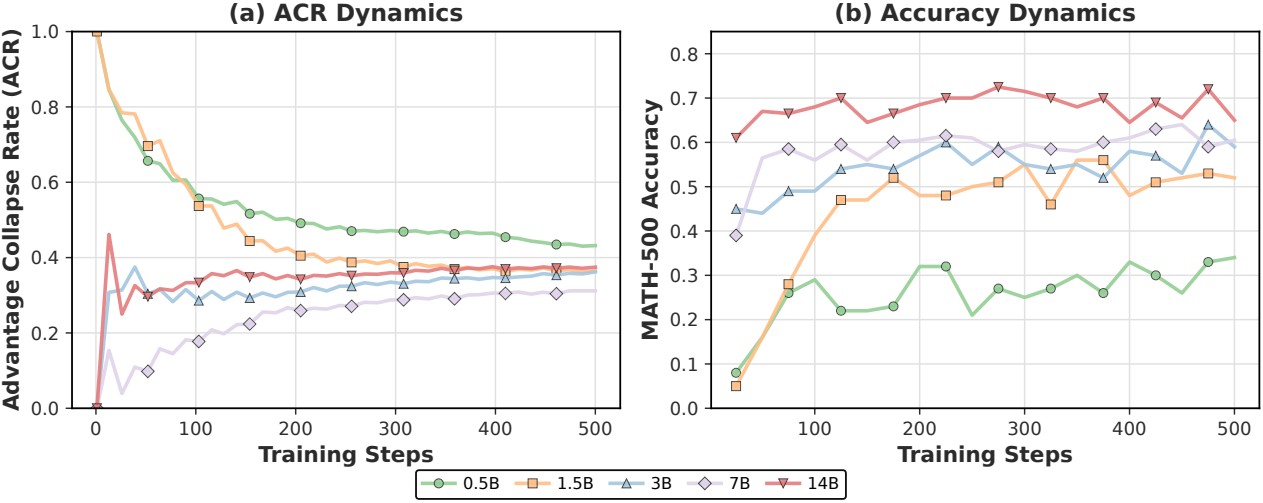

*Figure 6.* **ACR and accuracy training curves across model scales.** Left: ACR evolution over 500 training steps (orange shades). Larger models maintain consistently lower ACR. Right: Corresponding accuracy curves on MATH-500 (blue shades). The inverse relationship between ACR and accuracy is evident across all model scales.

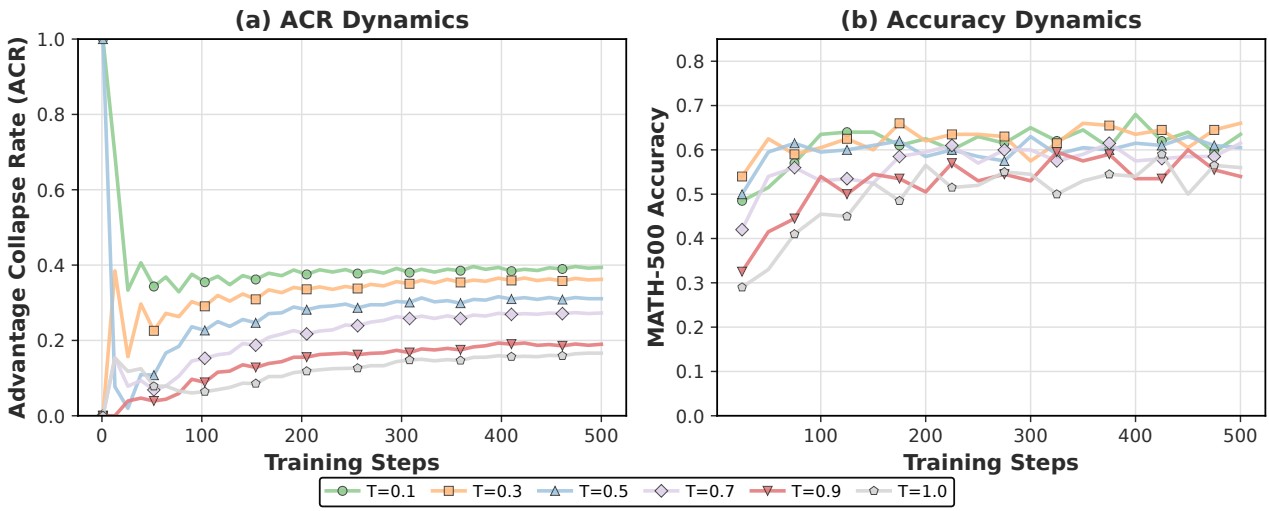

*Figure 7.* **ACR and accuracy training curves across sampling temperatures.** Left: ACR evolution shows strong temperature dependence (orange shades), with low temperatures causing persistent collapse. Right: Accuracy peaks at intermediate temperatures (blue shades) despite lower ACR at high temperatures.

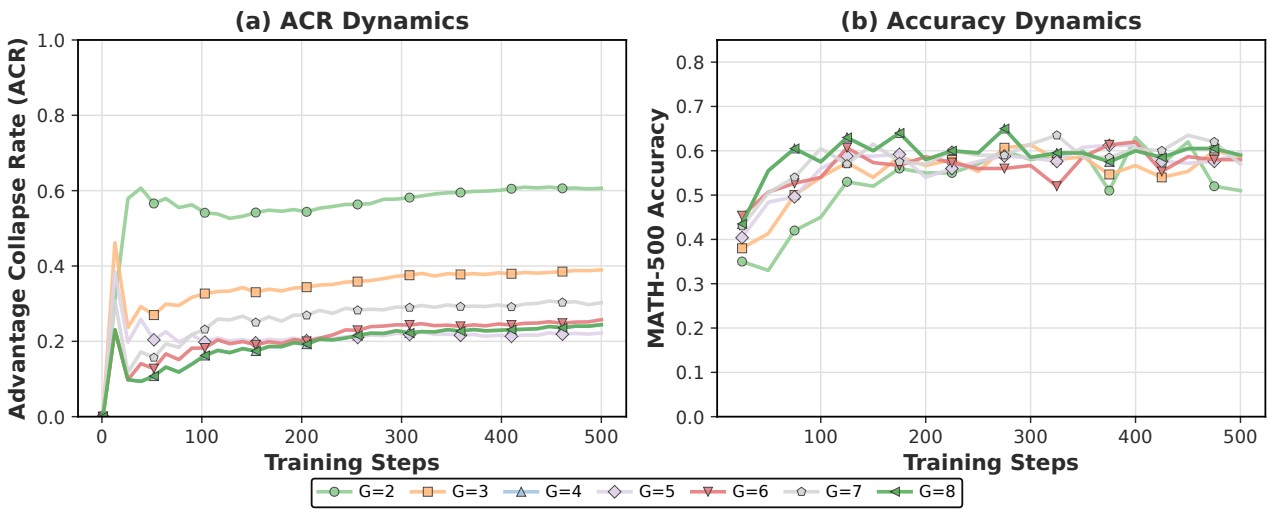

*Figure 8.* **ACR and accuracy training curves across group sizes.** Left: ACR dynamics (orange shades) showing that larger groups reduce collapse rate. Right: Accuracy dynamics (blue shades) with diminishing returns beyond $G = 6$.

### G.3. ACR Dynamics Across Group Sizes

Figure 8 shows how group size $G$ affects ACR dynamics:

- **Diminishing returns**: Increasing $G$ from 2 to 4 yields substantial ACR reduction, but further increases show diminishing benefits.

- **Variance reduction**: Larger groups produce smoother ACR curves due to statistical averaging effects.

### G.4. ACR Dynamics Across Dataset Difficulty

Figure 9 demonstrates the relationship between problem difficulty and ACR:

- **U-shaped collapse pattern**: Both very easy (Level 0–1) and very hard (Level 5–6) problems lead to high ACR, while intermediate difficulty (Level 3–4) maintains healthy reward diversity.

- **Difficulty-aware curriculum**: These results suggest that curriculum learning strategies selecting intermediate-difficulty problems could naturally mitigate advantage collapse.

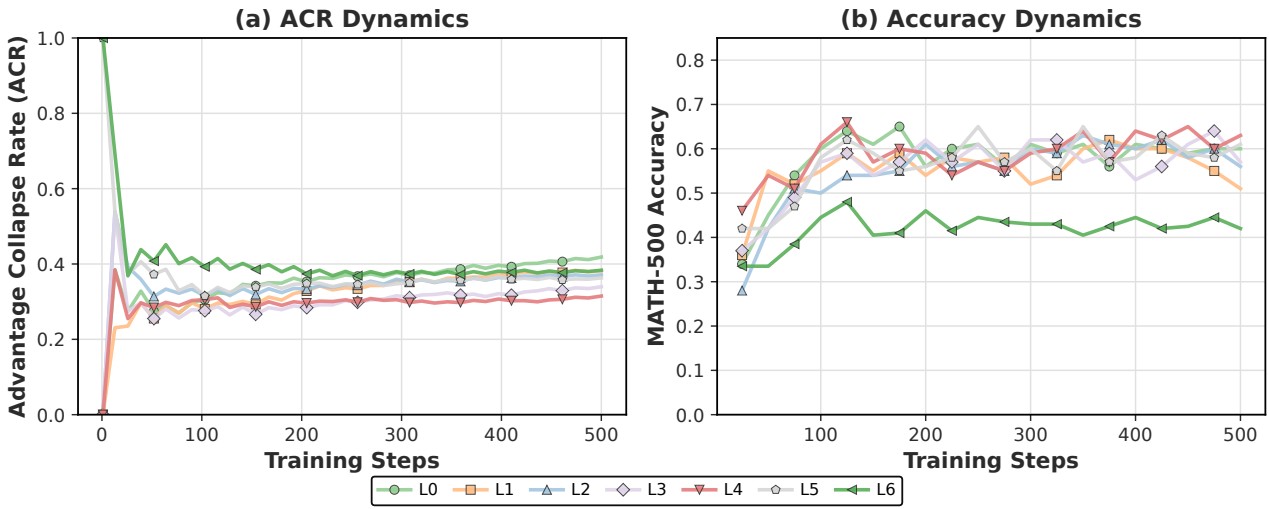

*Figure 9.* **ACR and accuracy training curves across dataset difficulty levels.** Left: ACR dynamics (orange shades) showing that both extremes (too easy or too hard) cause high collapse rates. Right: Accuracy dynamics (blue shades) with intermediate difficulty levels providing optimal learning conditions.

## H. Case Studies

To provide intuitive understanding of advantage collapse and the effectiveness of AVSPO, we present qualitative case studies using Qwen2.5-Math-1.5B, which exhibits the largest ACR reduction (0.40→0.15) among math-specialized models.

### H.1. Advantage Collapse in Training

Figure 11 illustrates two common scenarios where advantage collapse occurs during GRPO training. In the **all-incorrect** case (left), a challenging problem causes all $G = 8$ sampled responses to fail, yielding rewards $\mathcal{R} = \{0, 0, \ldots, 0\}$ with $\sigma_{\mathcal{R}} = 0$. In the **all-correct** case (right), an easier problem is solved by all responses, similarly producing $\sigma_{\mathcal{R}} = 0$. Both scenarios result in vanishing gradients under standard GRPO, wasting computational resources on ineffective training batches.

### H.2. Output Quality Comparison

Figure 10 presents example generations from models trained with GRPO versus AVSPO on representative MATH-500 problems. GRPO-trained models frequently fail to solve problems that AVSPO-trained models handle correctly. The reasoning chains from both models often begin with similar problem decomposition and initial steps, but GRPO gradually exhibits unjustified reasoning leaps, arithmetic errors, or premature guesses, whereas AVSPO maintains more rigorous logical progression toward the correct solution. This qualitative difference reflects AVSPO's ability to learn from a larger fraction of effective training batches by mitigating advantage collapse.

These case studies corroborate our quantitative findings: by mitigating advantage collapse, AVSPO enables more effective utilization of training signal, resulting in improved reasoning quality across diverse problem types.

**Question:** The variables $a$ and $b$ are inversely proportional. When the sum of $a$ and $b$ is 24, their difference is 6. What is $b$ when $a$ equals 5?

**Qwen-Math-1.5B-GRPO:**

Since (a) and (b) are inversely proportional, this means when one increases the other one also changes, so we can write the relationship as (a+b=k). From the question we know that (a+b=24) and also (a-b=6), so these two equations already give us enough information.

First we add the equations (a+b=24) and (a-b=6) together:
(a+b+a-b=24+6)
(a=30).

After finding (a), we substitute it back into the equation (a+b=24):
(30+b=24), therefore (b=-6).

Now we check the inverse proportionality condition. Since inverse proportion means the product is constant, we calculate (k) by multiplying (a) and (b):
(k=ab=30\times(-6)=-180). This confirms the relationship is correct.

Next, we are asked to find the value of (b) when (a=5). Using the same inverse proportion formula (ab=k), we substitute the values:
(5b=-180).
Solving this gives (b=-36).

Because the value is negative, we take the absolute value since (b) represents a quantity, so (b=36).

Therefore, when (a=5), the value of (b) is $\boxed{36}$ ⊗

**Qwen-Math-1.5B-AVSPO:**

Given that (a) and (b) are inversely proportional, we can write the relationship as (ab = k) for some constant (k). We are also given two conditions:

1. (a + b = 24)
2. (a - b = 6)

We can solve these two equations simultaneously to find the values of (a) and (b). First, add the two equations: ((a + b) + (a - b) = 24 + 6), which gives (2a = 30), so (a = 15).

Next, substitute (a = 15) into the first equation: (15 + b = 24), so (b = 9).

Now we know that when (a = 15), (b = 9). Since (a) and (b) are inversely proportional, we can find the constant (k) as (k = ab = 15 \cdot 9 = 135).

We need to find the value of (b) when (a = 5). Using the inverse proportionality relationship (ab = 135), substitute (a = 5): (5b = 135), so (b = \frac{135}{5} = 27).

Thus, the value of (b) when (a = 5) is = $\boxed{27}$. ✓

| ACR during training: 0.40 | ACR during training: 0.15 |

*Figure 10.* **Example generations from GRPO and AVSPO on MATH-500 problems.** Responses from Qwen2.5-Math-1.5B trained with GRPO (left) versus AVSPO (right) on the same problem. Both models start with similar reasoning approaches, but GRPO makes computational errors or reasoning leaps while AVSPO maintains logical consistency. Colors are added for visualization: blue indicates correct reasoning steps or answers, red indicates clearly incorrect steps, and orange indicates ambiguous steps that may lead to errors.

| **All-Incorrect Case** | **All-Correct Case** |
|---|---|
| **Question:** Find the sum of all integral values of $c$ with $c \le 25$ for which the equation $y=x^2-7x-c$ has two rational roots. | **Question:** The 3rd and 5th terms of an arithmetic sequence are 17 and 39, respectively. What is the 7th term of the same sequence? |
| **Response 1:** The initial config ... \boxed{-8}$. ✗ 
 **Response 2:** To solve the ... \boxed{-6}$. ✗ 
 ... 
 **Response 8:** I need to find all ... \boxed{26}. ✗ | **Response 1:** Let the first term ... \boxed{61}$. ✓ 
 **Response 2:** Let us solve the ... \boxed{61}$. ✓ 
 ... 
 **Response 8:** I need to find the ... \boxed{61}. ✓ |
| Rewards is: tensor([0., 0., 0., 0., 0., 0., 0., 0.]) 
 **$\sigma = 0$, ACR $= 1$** 
 ⚠ **Gradient $= 0$** | Rewards is: tensor([1., 1., 1., 1., 1., 1., 1., 1.]) 
 **$\sigma = 0$, ACR $= 1$** 
 ⚠ **Gradient $= 0$** |

*Figure 11.* **Advantage collapse examples during GRPO training. Left:** All-incorrect case where $G = 8$ responses fail on a challenging problem ($\sigma_{\mathcal{R}} = 0$, gradient vanishes). **Right:** All-correct case where all responses succeed ($\sigma_{\mathcal{R}} = 0$, gradient still vanishes). AVSPO addresses both scenarios by injecting virtual samples to restore reward variance.

## I. Statistical Significance Analysis

To assess the robustness of our reported improvements, we conduct multi-seed experiments on representative configurations spanning the full range of model scales. We select four models (Qwen2.5-0.5B, Qwen2.5-Math-1.5B, Qwen2.5-3B, and Qwen2.5-14B) covering 0.5B to 14B parameters, and two primary benchmarks (MATH-500 and GSM8K) for statistical validation.

*Table 9.* Multi-seed experimental results ($n = 5$ seeds) on representative configurations. We report mean $\pm$ standard deviation. All improvements are statistically significant under paired $t$-tests ($p < 0.05$).

| **Model** | **Benchmark** | **GRPO** | **AVSPO** | **$\Delta$** | **$p$-value** |
|---|---|---|---|---|---|
| Qwen2.5-0.5B | MATH-500 
 GSM8K | $24.6 \pm 1.9$ 
 $35.2 \pm 2.3$ | $31.4 \pm 1.6$ 
 $44.8 \pm 1.9$ | +6.8 
 +9.6 | 0.005 
 0.002 |
| Qwen2.5-Math-1.5B | MATH-500 
 GSM8K | $58.6 \pm 1.4$ 
 $49.8 \pm 1.8$ | $67.2 \pm 1.2$ 
 $59.3 \pm 1.5$ | +8.6 
 +9.5 | 0.003 
 0.006 |
| Qwen2.5-3B | MATH-500 
 GSM8K | $36.8 \pm 1.6$ 
 $52.6 \pm 2.1$ | $42.7 \pm 1.3$ 
 $61.3 \pm 1.7$ | +5.9 
 +8.7 | 0.012 
 0.008 |
| Qwen2.5-14B | MATH-500 
 GSM8K | $72.5 \pm 1.1$ 
 $71.8 \pm 1.4$ | $78.9 \pm 0.9$ 
 $77.4 \pm 1.1$ | +6.4 
 +5.6 | 0.001 
 0.004 |

Figure 12 visualizes the performance distributions, complementing the summary statistics in Table 9.

**Key Observations:**

- **Consistent improvements**: AVSPO outperforms GRPO across all seed-model-benchmark combinations, with gains ranging from +5.6 to +9.6 percentage points. As shown in Figure 12, the distributions are completely non-overlapping.

- **Statistical significance**: All eight comparisons yield $p < 0.05$ under paired $t$-tests, confirming that the observed improvements are unlikely due to random variation.

- **Lower variance**: AVSPO exhibits consistently lower standard deviation than GRPO (mean std: 1.40 vs. 1.70), suggesting that mitigating advantage collapse also stabilizes training outcomes. This is visually evident from the narrower boxplots for AVSPO in Figure 12.

- **Reproducibility of gains**: The multi-seed mean improvements closely match our single-run results reported in Table 1, validating the reliability of the main experimental findings across all model scales from 0.5B to 14B.

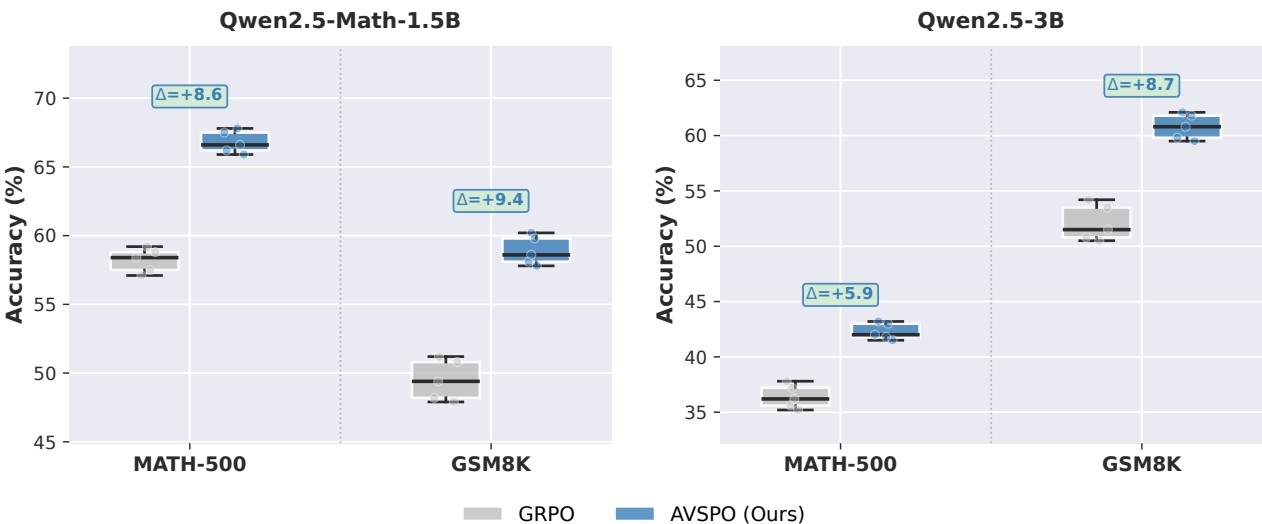

*Figure 12.* **Performance distribution across 5 random seeds.** Boxplots show median, quartiles, and individual data points for representative model-benchmark combinations. AVSPO (blue) consistently outperforms GRPO (gray) with non-overlapping distributions, visually confirming the statistical significance reported in Table 9. The narrower boxes for AVSPO indicate lower variance, suggesting more stable training.

