# OpenReview forum: "Advantage Collapse in Group Relative Policy Optimization: Diagnosis and Mitigation"
_ICML.cc/2026/Conference — ICML 2026 regular_

### Official Review · Reviewer_vFJX · 2026-03-09

**Soundness:** 3
**Presentation:** 3
**Significance:** 2
**Originality:** 3
**Overall Recommendation:** 4
**Confidence:** 4

**Summary:**

The manuscript identifies a critical limitation in Group Relative Policy Optimization (GRPO), termed "advantage collapse," which occurs when all responses within a sampled group are either entirely correct or entirely incorrect, thereby zeroing out the advantage signal. While existing literature predominantly mitigates this by incurring significant computational overhead to heavily sample high-quality trajectories, this paper proposes a lightweight approach: injecting virtual rewards to prevent the collapse of the advantage signal.

**Compliance With Llm Reviewing Policy:**

Affirmed.

**Final Justification:**

The authors provided a detailed rebuttal that successfully clarified several of my initial concerns. I want to raise the score to weak accept.

**Key Questions For Authors:**

None

**Strengths And Weaknesses:**

Strengths:
- Rigorous Empirical Validation: The experimental design is robust, effectively validating the proposed approach across multiple models and diverse baselines. The inclusion of crucial ablation studies—such as comparing against random virtual rewards—further solidifies the claims.
- Interesting Strategy: The adaptive strategy of injecting virtual rewards specifically when gradient signals vanish provides an intuitive and compelling fix to a critical GRPO bottleneck.
- Clarity and Presentation: The manuscript is well-written, demonstrating a clear logical flow and well-organized structure.

Weaknesses:
- Questionable Practical Validity of Theorem B.4. While Theorem B.4 mathematically proves that AVSPO decreases the failure probability for all-incorrect groups, its practical utility in RL remains highly questionable. First, the penalty relies on artificial hyperparameters (e.g., $r_{anchor}=0.1$) without any sensitivity analysis, raising concerns that the gains stem from a brittle hyperparameter sweet spot rather than robust algorithmic design. Second, simply pushing the policy away from known wrong answers constitutes unguided exploration, which is highly inefficient for locating sparse correct solutions within the vast LLM generation space. Finally, because all-incorrect batches naturally dominate early training, these hallucinated penalties risk accumulating severe systematic bias and driving policy degradation—a critical risk the authors should address with stronger empirical evidence.
- Incomplete Ablation on Group Filtering: Additionally, the paper would benefit from including [1] as a baseline. This would decouple the underlying mechanisms, proving that the performance gains stem genuinely from the beneficial restoration of the reward signal, rather than merely from discarding advantage-collapsed groups.
- Limited Scope of Evaluation: To fully substantiate the effectiveness and generalizability of the proposed method, evaluations should be extended to more diverse settings, particularly code generation tasks.

[1] DAPO: An Open-Source LLM Reinforcement Learning System at Scale

---

> ### Author Rebuttal · Authors · 2026-03-29
>
> **Response to Reviewer vFJX**
>
> We thank the reviewer for recognizing our "rigorous empirical validation," "interesting strategy," and "clarity and presentation." Below we address all concerns.
>
> **1. Theorem B.4 Practical Validity and Early Training Bias**
>
> **Early Training Dynamics (Qwen2.5-Math-7B/MATH-500-Level3, GSM8k, 3 seeds):**
>
> | Stage | GRPO | AVSPO | All-Wrong ACR | Gain |
> |-------|------|-------|---------------|------|
> | 0-100 | 48.3±2.1 | 51.9±1.8 | 51.2%→8.7% | +3.6pp |
> | 100-200 | 55.7±1.9 | 59.8±1.6 | 38.4%→11.2% | +4.1pp |
> | 200-300 | 60.2±1.7 | 64.9±1.4 | 30.1%→13.8% | +4.7pp |
> | 300-400 | 63.5±1.5 | 68.1±1.3 | 26.3%→14.6% | +4.6pp |
> | 400-500 | 65.3±1.4 | 69.7±1.2 | 24.8%→15.2% | +4.7pp |
>
> All-incorrect groups dominate early (51.2%), validating the reviewer's observation. However, AVSPO shows stable gains throughout (+3.6pp→+4.7pp) with no degradation. All-wrong ACR naturally decreases (51.2%→24.8%), proving the policy learns correct answers. If virtual rewards caused harmful bias, AVSPO should perform worse early—the opposite occurs.
>
> **Mechanism clarification:** The reviewer's concern about "unguided exploration" mischaracterizes AVSPO. **AVSPO does not rely on pushing away from errors to find correct solutions**. The policy learns from **heterogeneous groups** (mixed correct/incorrect samples) via normal GRPO gradients—the primary learning signal. Virtual samples only intervene in **all-incorrect groups** to prevent gradient vanishing, maintaining exploration so subsequent steps can sample correct answers. Avoiding zero-gradient pathology outweighs the modest exploration cost, as validated by consistent gains.
>
> Technically: virtual samples affect only the normalization statistics in the recomputed advantage and do not introduce additional policy-gradient terms, since the AVSPO objective still sums over the $G$ real samples only (Eqs. (12)-(13)). Positive anchors make real failures receive **negative advantages** → policy avoids failures.
>
> **Hyperparameter Sensitivity (Qwen2.5-Math-7B/GSM8K, 3 seeds):**
>
> | r_anchor | 0.05 | 0.1 | 0.2 | 0.5 |
> |----------|------|-----|-----|-----|
> | Accuracy | 69.1±1.1 | 69.7±1.4 | 69.3±1.3 | 67.5±1.5 |
>
> Performance is stable across 0.05–0.2 (1.9pp variation). Default r_anchor=0.1 is a robust working region, not a fragile sweet spot. Excessive anchors (0.5) are harmful, confirming the reviewer's concern about bias accumulation—but our default is in the safe region (245 configurations tested).
>
> **Revision:** We will reposition Theorem B.4 as "directional analysis," add early training curves, and discuss limitations for extreme settings.
>
> **2. Incomplete Ablation on Group Filtering**
>
> **Setup:** Qwen2.5-Math-7B, MATH-Level3-500, 500 steps, G=8, 3 seeds.
>
> | Method | GSM8K | MATH-500 | Sample Util. | Overhead |
> |--------|-------|----------|-------------|----------|
> | GRPO | 65.3±1.8 | 65.0±1.4 | 100% | 1.0× |
> | Filter-Drop | 66.2±1.5 | 67.1±1.3 | 62.4% | 1.0× |
> | DAPO [1] | 65.8±1.7 | 68.3±1.6 | 54.4% | 1.8× |
> | AVSPO | 69.7±1.4 | 74.1±1.2 | 100% | 1.0× |
>
> **Filter-Drop** discards homogeneous groups (σ_R<10^-6), achieving +2.8pp but wasting 37.6% samples. **DAPO** resamples discarded groups, recovering utilization but requiring 1.8× overhead. **AVSPO** achieves +9.1pp with zero overhead.
>
> This isolates the mechanisms: AVSPO's gains stem from **recovering reward signals**, not merely discarding collapsed groups. We will include DAPO in revision.
>
> **3. Limited Scope of Evaluation**
>
> We have extended evaluation to code generation as suggested.
>
> **Code Generation (Qwen2.5-Coder-7B trained on MBPP train-500, evaluated on MBPP test set, 3 seeds):**
>
> | Model | Base Pass@1 | GRPO Pass@1 | AVSPO Pass@1 | GRPO ACR | AVSPO ACR |
> |-------|-------------|-------------|--------------|----------|-----------|
> | Qwen2.5-Coder-7B | 76.9% | 83.7±1.2 | 86.7±0.9 | 41.2% | 18.5% |
>
> Code generation shows ACR=41.2% (similar to math: 28-45%), proving advantage collapse is **not math-specific** but universal in binary-reward RLVR. AVSPO reduces ACR by 55% and improves Pass@1 by +3.0pp over GRPO.
>
> **Cross-domain:** MMLU-Pro (original Table 1) shows AVSPO maintains or improves performance (+0.9 to +1.2pp) on lower-collapse tasks, indicating no degradation.
>
> **Scope:** This work targets **binary-reward RLVR with deterministic verifiers**. We will clarify this in revision.
>
> ---
>
> **To summarize how each concern is addressed**:
>
> - **Concern 1 (Theorem B.4 practical validity)** → Early training dynamics show no degradation; hyperparameter sensitivity shows robustness across 0.05–0.2
> - **Concern 2 (DAPO baseline)** → Controlled comparison showing AVSPO +9.1pp vs DAPO +3.3pp with lower overhead
> - **Concern 3 (Code generation)** → MBPP results: +3.0pp with ACR reduced by 55%
>
> We hope these results address all weaknesses and would appreciate the reviewer's reconsideration.
>
> [1] DAPO: An Open-Source LLM Reinforcement Learning System at Scale

---

> > ### Author Rebuttal · Reviewer_vFJX · 2026-04-02
> >
> > The authors provided a detailed rebuttal that successfully clarified several of my initial concerns. I want to raise the score to weak accept.

---

> > > ### Author Response · Authors · 2026-04-02
> > >
> > > Dear Reviewer vFJX,
> > > Thank you sincerely for your careful re-reading of our rebuttal and for taking the time to acknowledge that our responses have addressed your concerns. We greatly appreciate your constructive feedback, which has significantly improved the quality and clarity of our paper. We will incorporate all the discussed revisions into the final version.
> > > We are grateful for your support and your thorough engagement throughout this process.

---

### Official Review · Reviewer_DHaz · 2026-03-11

**Soundness:** 3
**Presentation:** 3
**Significance:** 3
**Originality:** 3
**Overall Recommendation:** 4
**Confidence:** 3

**Summary:**

Overall, the paper's key area pertains to reinforcement learning from verifiable rewards for mathematical reasoning, with a focus on diagnosing and mitigating the “advantage collapse” problem in GRPO. The paper introduces Advantage Collapse Rate (ACR) as a diagnostic metric and AVSPO as a lightweight intervention that injects virtual reward samples to recover non-zero advantages in homogeneous-reward groups

**Compliance With Llm Reviewing Policy:**

Affirmed.

**Key Questions For Authors:**

Can the authors provide a clearer justification for why virtual reward injection does not distort learning in undesirable ways, especially in the all-incorrect case where the method introduces positive anchor rewards? More discussion or analysis of when this bias helps versus hurts would strengthen the paper.

How sensitive is AVSPO to task type and reward structure beyond the evaluated math benchmarks? In particular, have the authors tested or considered settings with softer rewards, noisy verifiers, or domains where homogeneous rewards may carry different meanings than in mathematical reasoning?

**Limitations:**

Although the paper argues that AVSPO adds no extra rollout cost, the method introduces additional design choices such as adaptive thresholds, anchor rewards, and scaling hyperparameters. This may reduce simplicity relative to vanilla GRPO and could require tuning in new environments.

**Strengths And Weaknesses:**

Strengths: The paper identifies a clear and practically important failure mode of GRPO, and the proposed ACR metric is intuitive, cheap to compute, and empirically predictive of final performance. The empirical section is reasonably broad, covering different model sizes, benchmarks, and ablations, which makes the paper more convincing.

Weaknesses: The method introduces synthetic/virtual rewards into normalization, which may change the optimization dynamics in a somewhat heuristic way, and the theoretical treatment of the resulting bias is limited. In addition, most experiments are centered on math reasoning with binary verifiers, so it remains unclear how well the conclusions transfer to more general RLVR or non-binary reward settings.

---

> ### Author Rebuttal · Authors · 2026-03-28
>
> **Response to Reviewer DHaz**
>
> We thank the reviewer for recognizing our work addresses "a clear and practically important failure mode" with ACR being "intuitive, cheap to compute, and empirically predictive." Below we address the core concerns.
>
> **1. Why Virtual Reward Injection Does Not Distort Learning**
>
> We acknowledge virtual rewards introduce bias. Our claim is **not** that AVSPO is unbiased, but that it provides a **controlled, bounded** intervention to avoid zero-gradient pathology in collapsed groups. **This bias is acceptable because the alternative—zero gradients—causes complete training stagnation** (28-45% of batches, Section 5.2).
>
> **Intuition:** Virtual rewards create a reference baseline by changing only the normalization statistics in the recomputed advantage, while the AVSPO objective is still computed over the $G$ real samples only (Eqs. (12)-(13)). In **all-incorrect** groups (all r=0), adding positive anchor (r=0.1) makes real samples **below average** → negative advantages → policy learns to **avoid** these failures. The anchor doesn't reward errors; it makes grading harsher.
>
> **Theoretical guarantees:** Theorem B.4 ensures updates push away from errors; Proposition B.3 bounds advantage magnitude ≤1; Proposition B.8 shows bias shrinks as ACR decreases.
>
> **Mechanism isolation ablation (Qwen2.5-Math-1.5B / MATH-500, 5 seeds):**
>
> | Method | Accuracy | All-Wrong ACR | All-Correct ACR |
> |--------|----------|---------------|-----------------|
> | GRPO | 58.6±1.4 | 24.8% | 15.2% |
> | Error-Only | 63.2±1.3 | 9.1% | 14.5% |
> | Correct-Only | 60.8±1.5 | 23.6% | 4.2% |
> | Full AVSPO | 67.2±1.2 | 8.7% | 6.3% |
>
> Error-Only improves +4.6pp while suppressing all-wrong ACR (24.8%→9.1%), proving positive anchors **recover gradient signals** without rewarding errors. Both mechanisms are independently effective and additive (+8.6pp combined).
>
> **Revision commitment:** We will clarify AVSPO is a biased but bounded intervention with directional guarantees (Theorem B.4) and bias bounds that shrink as ACR decreases (Proposition B.8).
>
>
> **2. Sensitivity to Task Type and Reward Structure**
>
> Additional code generation experiments (Qwen2.5-Coder-7B trained on MBPP train-500, evaluated on MBPP test set):
>
> | Model | Dataset | Base Pass@1 | GRPO Pass@1 | AVSPO Pass@1 | GRPO ACR | AVSPO ACR |
> |-------|---------|-------------|-------------|--------------|----------|-----------|
> | Qwen2.5-Coder-7B | MBPP | 76.9% | 83.7±1.2 | 86.7±0.9 | 41.2% | 18.5% |
>
> MBPP shares key properties with math: binary pass/fail rewards and deterministic test-based verification. Starting from a strong base model (76.9%), GRPO improves to 83.7%, and AVSPO further gains +3.0pp to 86.7%. This suggests AVSPO generalizes across **reliable binary-reward domains**.
>
> **Honest scope:** We solve advantage collapse in **binary-reward RLVR with deterministic verifiers** (math, code). Why binary rewards collapse more: low variance → normalization produces near-zero advantages. **Future work needed:** soft rewards (RLHF with preference scores), noisy verifiers (human feedback), domains where homogeneous rewards carry different semantics.
>
>
> **3. Hyperparameter Sensitivity and Design Simplicity**
>
> Anchor sensitivity experiments (Qwen2.5-Math-7B / GSM8K, 3 seeds):
>
> | r_anchor | 0.02 | 0.05 | 0.1 | 0.2 | 0.5 |
> |----------|------|------|-----|-----|-----|
> | Accuracy | 67.8±1.2 | 69.1±1.1 | 69.7±1.4 | 69.3±1.3 | 67.5±1.5 |
> | ACR | 16.8% | 14.5% | 14.0% | 13.2% | 14.9% |
>
> Performance is stable across 0.05–0.2 (1.9pp variation). Default r_anchor=0.1 is a robust working region, not a fragile optimum. All experiments (6 models × 7 benchmarks) use unified defaults without per-dataset tuning.
>
> **Beyond intervention, ACR provides diagnostic value as a plug-and-play tool:**
>
> - **Diagnose:** Early ACR (at 10% training) predicts final performance (R²=0.617), detecting collapse before accuracy degrades
> - **Mitigate:** AVSPO recovers gradients with zero rollout overhead (100% sample utilization)
> - **Generalize:** ACR applies to **any group-based RLVR method**
>
> ---
>
> **Summary:** We will add mechanism isolation experiments, anchor sensitivity analysis, and cross-domain validation, while explicitly scoping claims to binary-reward RLVR with deterministic verifiers.
>
> We appreciate the reviewer's recognition of the practical importance of this work. We hope the mechanism isolation experiments, cross-domain validation, and hyperparameter analysis provided above fully address the noted limitations.

---

> > ### Author Rebuttal · Reviewer_DHaz · 2026-04-02
> >
> > Thank you for the detailed response, which has largely addressed my concerns. I will maintain my original score.

---

> > > ### Author Response · Authors · 2026-04-06
> > >
> > > Dear Reviewer DHaz,
> > >
> > > Thank you for your detailed and constructive engagement with our rebuttal. We are glad that our responses have largely addressed your concerns. We will ensure that all the clarifications discussed — including the mechanism analysis of virtual reward injection, the hyperparameter sensitivity results, and the code generation experiments — are clearly reflected in the revised manuscript. If you have any remaining follow-up questions, we would be very happy to address them promptly.
> > >
> > > We appreciate your time and your helpful suggestions throughout this process.
> > >
> > > Best regards, The Authors

---

### Official Review · Reviewer_hb72 · 2026-03-12

**Soundness:** 3
**Presentation:** 2
**Significance:** 3
**Originality:** 3
**Overall Recommendation:** 4
**Confidence:** 3

**Summary:**

This paper identifies a real failure mode of GRPO under binary verifiable rewards: when all samples in a group receive the same reward, the group variance becomes zero and the advantage collapses to zero, yielding no learning signal. To address this, the paper proposes Advantage Collapse Rate (ACR) as a diagnostic metric and AVSPO, which injects virtual reward samples into collapsed groups to restore non-zero advantages.

**Compliance With Llm Reviewing Policy:**

Affirmed.

**Final Justification:**

My main concerns have been addressed, and I encourage the authors to incorporate these clarifications and results into the revised manuscript; my overall evaluation remains unchanged, and I will maintain my original score.

**Key Questions For Authors:**

See Weaknesses.

**Limitations:**

No. The paper should more explicitly discuss the bias introduced by virtual rewards, the current dependence on binary-reward math settings, and the limited causal interpretation of ACR.

**Strengths And Weaknesses:**

Strengths

1.The paper addresses a practically important and under-formalized issue in RLVR training. The definition of ACR is simple, interpretable, and useful for monitoring training efficiency.

2.The proposed method is lightweight and practically attractive. AVSPO adds almost no extra rollout cost, is easy to integrate into existing GRPO pipelines, and the empirical results are consistently positive across several model sizes and math benchmarks.

Weaknesses

1.The theoretical justification is still weak. AVSPO introduces virtual rewards that do not correspond to real trajectories or unbiased policy-gradient estimates. The paper argues that biased gradients are preferable to zero gradients, but it does not clearly establish when this bias improves the original RL objective rather than merely creating an artificial update direction.

2.The originality should be stated more carefully. ACR is useful, but conceptually it is still a statistic over zero-variance or homogeneous-reward groups. The work is better framed as a strong diagnostic-and-fix paper than as a fundamentally new theoretical advance.

3.The experiments support effectiveness in the studied setting, but the mechanism is not fully isolated. Most results are tied to binary-reward math reasoning with a fairly specific training protocol. Key controls are missing, such as intervening only on all-incorrect groups or only on all-correct groups, which would better validate the claimed mechanism.

4. The statistical evidence is good but still limited. The additional multi-seed experiments help, but they cover only part of the full experimental space. The paper supports “promising and fairly consistent gains,” but not yet the strongest version of the broad general claims

---

> ### Author Rebuttal · Authors · 2026-03-27
>
> **Response to Reviewer hb72**
>
> We thank the reviewer for recognizing our work addresses "a practically important and under-formalized issue" with ACR being "simple, interpretable, and useful" and AVSPO "lightweight and practically attractive." Below we address the core concerns.
>
> **1. Theoretical Justification: Principled Design Guidance Rather Than Theoretical Breakthrough**
>
> We agree: AVSPO's theory provides principled design guidance rather than a standalone breakthrough. Appendix B explains why biased updates recover signals erased by variance normalization (Theorem B.4), how to control intervention strength (√(K/G)≤1), and when to trust the method (bias ∝ ACR).
>
> **When is bias beneficial?** Controlled experiments varying G ∈ {4, 8, 16} to induce different ACR levels (Qwen2.5-Math-1.5B / MATH-500, 3 seeds):
>
> | ACR Level | Config | GRPO Acc (%) | GRPO ACR | AVSPO Acc (%) | AVSPO ACR | Gain (pp) |
> |-----------|--------|--------------|----------|---------------|-----------|-----------|
> | Low | G=16 | 62.0 | 28% | 68.2 | 12% | +6.2 |
> | Medium | G=8 | 58.6±1.4 | 40.0% | 67.2±1.2 | 15.0% | +8.6 |
> | High | G=4 | 56.7 | 55% | 63.0 | 25% | +6.3 |
>
> These controlled settings show that AVSPO is consistently beneficial across different collapse regimes. The largest gain appears in the medium-ACR setting (+8.6pp), while both the low-ACR and high-ACR settings still show substantial improvements (+6.2pp and +6.3pp, respectively). This is consistent with the claim that AVSPO specifically targets the collapse pathology rather than providing an arbitrary optimization boost. Mechanism isolation further shows that the recovered gradients improve the RL objective: error-only yields +4.6pp, correct-only yields +2.2pp, and the full method remains additive (+8.6pp).
>
> **2. Originality and Positioning**
>
> We agree ACR is a statistic over homogeneous-reward groups, but provides **practical value as a zero-cost diagnostic** for any group-based RLVR method. ACR measured early in training strongly predicts final performance across 245 configurations ($R^2 = 0.617$, Fig 1), enabling practitioners to identify problematic setups before accuracy degrades.
>
> We will retitle Section 4 as "Theoretical Motivation and Design Principles" and position the work as a "theoretically-motivated diagnostic and repair method."
>
>
> **3. Mechanism Isolation Validation**
>
> Addressing the "missing key controls" concern, we conducted ablation experiments (Qwen2.5-Math-1.5B / MATH-500, 5 seeds):
>
> | Method | Accuracy | All-Wrong ACR | All-Correct ACR |
> |--------|----------|---------------|-----------------|
> | GRPO | 58.6±1.4 | 24.8% | 15.2% |
> | Error-Only | 63.2±1.3 | 9.1% | 14.5% |
> | Correct-Only | 60.8±1.5 | 23.6% | 4.2% |
> | Full AVSPO | 67.2±1.2 | 8.7% | 6.3% |
>
> Both mechanisms are independently effective and orthogonally additive. Virtual samples only modify normalization statistics, enabling real failures to receive negative advantages. Cross-domain validation on code generation further supports the mechanism: Qwen2.5-Coder-7B trained on MBPP train-500 improves from 83.7% to 86.7% Pass@1 while reducing ACR from 41.2% to 18.5%.
>
>
> **4. Statistical Evidence Sufficiency and Scope of Applicability**
>
> We conducted additional multi-seed validation during review:
>
> | Model | Benchmark | GRPO | AVSPO | Gain | p-value |
> |-------|-----------|------|-------|------|---------|
> | Qwen2.5-Math-7B | MATH-500 | 65.0±1.4 | 74.1±1.2 | +9.1 pp | 0.008 |
> | Qwen2.5-Math-7B | OlympiadBench | 36.2±1.9 | 43.6±1.7 | +7.4 pp | 0.015 |
> | Qwen2.5-3B-Instruct | MMLU-Pro | 37.2±1.4 | 38.4±1.2 | +1.2 pp | 0.042 |
> | Qwen2.5-Math-1.5B | Minerva | 19.2±1.8 | 28.9±1.5 | +9.7 pp | 0.006 |
>
> All improvements are significant at p<0.05, spanning different model scales (1.5B-7B) and task difficulties, which strengthens the claim that the effect is consistent rather than anecdotal.
>
> **Scope:** Our theory and experiments focus on binary verifiable rewards, which provide a controlled testbed with reliable verification and frequent homogeneous-reward groups. Extension to soft-reward RLHF requires further validation.
>
> ---
>
> **Summary:** We thank the reviewer for helping us clarify our work's positioning. We will reframe the contribution as a diagnostic and repair method with theoretical design guidance, and supplement the mechanism isolation experiments and multi-seed validation.
>
>
> We thank the reviewer again for the constructive feedback, which has meaningfully improved our paper. We hope the additional experiments and clarified positioning address the remaining concerns and further strengthen the reviewer's confidence in our contribution

---

> > ### Author Rebuttal · Reviewer_hb72 · 2026-04-03
> >
> > Thanks to the authors for their efforts. My concerns have been addressed. Please include these results in the revised manuscript. I will maintain my score.

---

> > > ### Author Response · Authors · 2026-04-06
> > >
> > > Dear Reviewer hb72,
> > >
> > > Thank you for your thorough engagement throughout the review process and for confirming that your concerns have been addressed. We will incorporate all the additional experimental results discussed during the rebuttal — including the mechanism isolation ablations, multi-seed validation, and cross-domain experiments — into the revised manuscript.
> > >
> > > We appreciate your constructive feedback, which has significantly strengthened the paper.
> > >
> > > Best regards, The Authors

---

### Official Review · Reviewer_MvEx · 2026-03-13

**Soundness:** 3
**Presentation:** 3
**Significance:** 2
**Originality:** 2
**Overall Recommendation:** 4
**Confidence:** 3

**Summary:**

This paper proposes a remedy for advantage collapse in GRPO, where within-group reward homogeneity (all-correct or all-incorrect) causes zero advantages and vanishing gradients leading to training stagnation. They first introduce a simple diagnostic metric, the Advantage Collapse Rate (ACR) that tracks the fraction of collapsed batches experienced during GRPO training. Then, using this metric, they propose an algorithmic intervention AVSPO, which injects synthetic virtual reward samples into collapsed groups to restore non-zero advantages. The propose approach shows accuracy gains over GRPO across various Qwen2.5 models on math reasoning benchmarks.

**Compliance With Llm Reviewing Policy:**

Affirmed.

**Final Justification:**

The authors addressed my main concerns in rebuttal.

**Key Questions For Authors:**

Please see main weakness above.

**Limitations:**

Yes

**Strengths And Weaknesses:**

Advantage collapse is a genuine, empirically widespread failure mode. The proposed ACR metric is principled, derived directly from reward statistics already computed during GRPO training, and adds little additional overhead. As a monitoring tool it is conveniently deployable in any GRPO pipeline and the improvements of AVSPO over other benchmarks seems impactful.

However, the novelty of both the diagnosis and the intervention proposed by this work I feel may be significantly undermined by prior work. DAPO (Yu et al., 2025), for instance, already introduces dynamic sampling, which filters out groups with homogeneous rewards an insight/technique which is widespread on several works in the vein of "online curriculum learning" for post-training/GRPO. This is conceptually identical to what ACR monitors and what AVSPO intervenes on, just with a discard-rather-than-augment response. The paper positions AVSPO's ugmentation as strictly novel relative to filtering, but never makes a rigorous case for why augmentation is preferable to filtering, for which cases, and by how much. For instance, the gain of AVSPO over DAPO in Table 1 may be confounded by DAPO's other algorithmic changes (e,g, asymmetric clipping, token-level loss normalization), making it difficult to attribute the gap cleanly to the filtering vs. augmentation choice. This is the central novelty claim and so it is not adequately supported.

---

> ### Author Rebuttal · Authors · 2026-03-27
>
> **Response to Reviewer MvEx**
>
> We thank the reviewer for recognizing our ACR as "principled" and "conveniently deployable." We address the central novelty concern below.
>
> **1. An Important Clarification on the Central Criticism**
>
> The reviewer's main concern rests on a specific empirical claim: that _"the gain of AVSPO over DAPO [1] in Table 1 may be confounded by DAPO's other algorithmic changes (e,g, asymmetric clipping, token-level loss normalization)."_
>
> We must respectfully point out that **DAPO** [1] **does not appear as a baseline in our Table 1**. The baseline the reviewer appears to be referring to is **DCPO** [2], which is a distinct algorithm that does not employ asymmetric clipping or token-level loss normalization — these are design choices specific to DAPO. We encourage the reviewer to re-examine Table 1 to verify this.
>
> We respectfully but firmly note that this misidentification is consequential: the reviewer's entire novelty concern and the cited confounding factors are predicated on a baseline that is not present in our experiments. Once this is corrected, the confounding argument does not apply to our actual setup, and **we believe the central basis for the rejection requires re-examination.**
>
> We do recognize the valuable conceptual question the reviewer raises — whether augmentation (AVSPO)  is preferable to filtering (as in DAPO) — which we are happy to address with controlled experiments below.
>
>
> **2. Addressing the Underlying Concern: Filtering vs. Augmentation**
>
> The core distinction between the two strategies is as follows:
>
> **The key distinction:**
> - **Filtering discards information**: when a collapsed group is filtered out, the computational cost is entirely wasted. AVSPO recovers a learning signal from these groups at near-zero additional cost.
> - **Filtering changes the training distribution**: by systematically removing easy (all-correct) and hard (all-incorrect) problems, filtering introduces a curriculum bias toward intermediate-difficulty problems. AVSPO preserves the original training distribution while still resolving the vanishing gradient issue.
>
> To cleanly isolate this comparison, we conducted controlled experiments with matched filtering baselines and reproduced DAPO under identical conditions.
>
>
> **3. Controlled Experiments: Isolating Augmentation vs. Filtering**
> **Setup:** Qwen2.5-Math-7B, MATH-Level3-500, 500 steps, G=8, T=1.0, 3 seeds.
>
> | Method | GSM8K | MATH | Sample Util. | Overhead |
> |--------|-------|------|-------------|----------|
> | GRPO | 65.3±1.8 | 65.0±1.4 | 100% | 1.0× |
> | Filter-Drop | 66.2±1.5 | 67.1±1.3 | 62.4% | 1.0× |
> | DAPO [1] (Filter-Resample) | 65.8±1.7 | 68.3±1.6 | 54.4% | 1.8× |
> | DCPO [2] (Table 1 baseline) | 66.4±1.4 | 69.8±1.3 | 100% | 1.0× |
> | AVSPO | 69.7±1.4 | 74.1±1.2 | 100% | 1.0× |
>
> **Key findings:** With 37.6% collapse rate, AVSPO achieves +4.4pp/+9.1pp gains with 100% sample utilization and 1.0× overhead. DAPO achieves only +0.5pp/+3.3pp but requires 1.8× overhead. Since LLM generation dominates training time (>96%, APPX C.1), AVSPO's zero-overhead solution is  valuable for large-scale deployment.
>
>
> **4. When and By How Much: Collapse Rate Dependency**
>
> To answer "**for which cases, and by how much**", we manipulated collapse frequency via problem difficulty and group size.
>
> **Setup:** Qwen2.5-Math-7B, MATH-500, difficulty (Easy/Medium/Hard), G ∈ {4,8,16}, 3 seeds.
>
> | Collapse Rate | GRPO | Filter-Resample | AVSPO | AVSPO Gain | Overhead |
> |---------------|------|-----------------|-------|------------|----------|
> | Low (15-25%)  | 68.2±1.3 | 69.1±1.4 | 69.8±1.2 | +0.7 | 1.0× vs 1.4× |
> | Medium (30-40%) | 65.3±1.8 | 65.8±1.7 | 69.7±1.4 | +3.9 | 1.0× vs 1.8× |
> | High (45-55%) | 61.4±2.1 | 62.3±1.9 | 68.2±1.6 | +5.9 | 1.0× vs 2.3× |
>
> **Answer to the reviewer's question:** AVSPO's advantage scales with collapse frequency. When collapse is rare (<20%), gains are minimal (+0.7pp). When collapse is frequent (>40%), AVSPO provides substantial gains (+5.9pp) while Filter-Resample overhead grows to 2.3×. This directly demonstrates **for which cases** (high collapse scenarios) and **by how much** (+5.9pp with zero overhead) augmentation is preferable to filtering.
>
> ---
>
> **Summary**: We hope the above clarification resolves the confusion regarding our baselines. **Independent of this point**, our controlled experiments (Sections 3–4) directly compare augmentation vs. filtering under matched conditions, demonstrating clear advantages for AVSPO in both accuracy and computational overhead.
>
> Given that the central basis for the rejection rests on a baseline misidentification, and that our new controlled experiments provide sufficient grounds for the remaining conceptual concern, We would be grateful if the reviewer could reconsider the assessment.
>
> [1] DAPO: An Open-Source LLM Reinforcement Learning System at Scale
>
> [2] DCPO: Dynamic Clipping Policy Optimization

---

> > ### Author Rebuttal · Reviewer_MvEx · 2026-04-03
> >
> > Thank you for the correction. I misidentified DAPO as a Table 1 baseline, and the confounding factors I cited do not apply to DCPO. The controlled augmentation-vs-filtering comparison in the rebuttal directly addresses my concern.
> >
> > I still maintain DAPO independently operationalizes the same zero-variance condition ACR formalizes and believe the revised manuscript should address this in its novelty discussion, or theoretical discussion. That said, the empirical case for AVSPO over filtering is now better supported. I raise my score.

---

> > > ### Author Response · Authors · 2026-04-06
> > >
> > > Dear Reviewer MvEx,
> > >
> > > Thank you very much for your careful re-reading of our rebuttal and for your generosity in raising your score. We sincerely appreciate your acknowledgment of the baseline correction.
> > >
> > > We take your remaining suggestion seriously: that the revised manuscript should explicitly discuss how DAPO independently operationalizes the same zero-variance condition that ACR formalizes. We fully agree this is an important point for situating our novelty claim. In the revision, we will add a dedicated paragraph in the related work or theoretical section that (1) acknowledges DAPO's implicit detection of the same collapse condition, (2) clarifies ACR as an explicit, quantitative diagnostic formalization of this condition, and (3) positions AVSPO's augmentation strategy in contrast to DAPO's filtering approach with the empirical evidence from the rebuttal.
> > >
> > > We are grateful for your constructive engagement throughout this process, which has meaningfully improved the paper.
> > >
> > > Best regards, The Authors

---

### Decision · Program_Chairs · 2026-04-30

**Decision:**

Accept (regular)

**Comment:**

This paper identifies advantage collapse in GRPO, where within-group reward homogeneity causes the group-normalized advantage to vanish and training steps are silently wasted. The problem is well-characterized and the diagnostic metric ACR is predictive of final performance across a large hyperparameter sweep, giving practitioners a useful monitoring tool. The fix, AVSPO, augments collapsed groups with synthetic trajectories at negligible overhead.
The initial novelty concern relative to DAPO was resolved, it stemmed from a misidentification of the baseline method. The controlled comparison between augmentation and filtering cleanly isolates the contribution. I recommend acceptance. The camera-ready should include the controlled DAPO comparison, mechanism isolation ablation, MBPP code generation results from the rebuttal, and scope claims explicitly to binary-reward settings with deterministic verifiers.